# Observation of multi-order polar radial vortices and their topological transition

Wan-Rong Geng[1,8], Xiangwei Guo [2,8], Yin-Lian Zhu [1,3], Desheng Ma [4], Yun-Long Tang [5], Yu-Jia Wang [5], Yongjun Wu[2], Zijian Hong [2] ✉ & Xiu-Liang Ma [1,6,7] ✉

Topological states have garnered enormous interest in both magnetic and ferroelectric materials for promising candidates of next-generation information carriers. Especially, multi-order topological structures with modulative topological charges are promising for multi-state storage. Here, by engineering boundary conditions, we directly observe the self-assembly two-order ferroelectric radial vortices in high-density $BiFeO_3$ nanostructures. The as-observed two-order radial vortex features a doughnut-like out-of-plane polarization distribution and four-quadrant in-plane distribution, with the topological charge of $Q = 0$. Systematic dimensional control of the $BiFeO_3$ nanostructures reveals size-dependent stabilization of distinct topological states, from elementary one-order to complex three-order radial vortices, which is further rationalized by phase-field simulations. The transition between different topological states with various topological charges is also realized under an external electric field. This study opens up an avenue for generating configurable polar topological states, offering potential advancements in designing high-performance multi-state memory devices.

The configurable spin topological defects in magnetic materials have been proven to be the source of many exotic phenomena with potential applications in electronic devices[1–3]. Complex topological structures are generally characterized by their topological charge $Q$, defined as a measure of the wrapping of spin vectors around a unit sphere, which governs the stability and dynamics of the topological states[4–6]. While numerous topological textures have been extensively studied, the majority exhibit topological charges confined to $|Q| \leq 1$. For example, (anti)skyrmion, known as the nanoscale spin vortex characterized by non-trivial real-space topological configuration[2,5], stands as the prototypical topological state with $Q = \pm 1$. It has been reported that this topological entity can undergo fractionalization into (anti)meron pairs bearing the fractional topological charge of $Q = \pm \frac{1}{2}$

(ref. [7]). Besides, the (anti)vortex is another kind of ubiquitous topological defects with $Q = \pm 1$ (ref. [7]). These spin structures display a variety of exotic characteristics, including robust topological protection[7] and self-organized lattice ordering[8,9].

Based on these topological sates, the modulation of topological charges enables the promising applications in high-density memory and neuromorphic computing[10,11]. Especially, the stabilization of the skyrmion bundles, the multi-$Q$ three-dimensional skyrmionic textures, realizes the precise tuning of collective topological charges reaching $Q = 55$ (ref. [12]). Similarly, the family of target skyrmion consisting of a central skyrmion surrounded by one or more concentric helical stripes[13] could also realize the alternative topological charges. As one intriguing member of the target skyrmion family[13], skyrmionium, with

[1]Bay Area Center for Electron Microscopy, Songshan Lake Materials Laboratory, Dongguan 523808, China. [2]State Key Laboratory of Silicon and Advanced Semiconductor Materials, School of Materials Science and Engineering, Zhejiang University, Hangzhou 310058, China. [3]Hunan University of Science and Technology, Xiangtan 411201, China. [4]School of Applied and Engineering Physics, Cornell University, Ithaca, NY 14850, USA. [5]Shenyang National Laboratory for Materials Science, Institute of Metal Research, Chinese Academy of Sciences, Shenyang 110016, China. [6]Institute of Physics, Chinese Academy of Sciences, Beijing 100190, China. [7]Quantum Science Center of Guangdong-HongKong-Macau Greater Bay Area (Guangdong), Shenzhen 510290, China. [8]These authors contributed equally: Wan-Rong Geng, Xiangwei Guo. ✉e-mail: hongzijian100@zju.edu.cn; xlma@iphy.ac.cn

the doughnut-like out-of-plane (OOP) spin texture[14–17], exhibits a vanishing net topological charge ($Q = 0$), largely suppressing the detrimental effect of skyrmion Hall effect[14].

While magnetic topological states and the control of topological charges have been extensively studied, the emergent topological domains in ferroelectrics have recently garnered significant attention as their promising applications for high-density data storage[18], as well as ultralow power negative capacitance field effect transistors[19]. By elaborately regulating the interplay of elastic, electrostatic and gradient energies in the dedicated ferroelectric systems, several complex topological states and topological orderings could be stabilized[20–30]. For example, the vortices, orderly stabilized in PbTiO$_3$/SrTiO$_3$ superlattices[21], display interesting functionalities distinct from the bulk domains, such as negative capacitance[31], emergent chirality[32] and special subterahertz collective mode-vortexon[33]. Furthermore, the thermal responses of polar vortices exhibit a wealth of intermediate states thereby triggering multi-state switching of ferroelectric topologies[34]. Despite the remarkable progress in ferroelectrics, the existing studies predominately focus on the low-order polar topological structures, leaving the high-order configurations with tunable topological charges largely unexplored.

Herein, the stabilization and modulation of ferroelectric two-order radial vortex and the topological transition between multi-order vortices are observed in high-density self-assembly BiFeO$_3$ (BFO) nanostructures. Piezoresponse force microscopic (PFM) and aberration-corrected scanning transmission electron microscopic observations indicate that the polarization configuration of ferroelectric two-order radial vortex with the topological charge of $Q = 0$ features a doughnut-like OOP texture and four-quadrant in-plane (IP) distribution. Furthermore, multi-order radial vortices with different topological charges, including one-order radial vortex and three-order radial vortex, can also be stabilized by tuning the sizes of BFO nanostructures, confirmed by phase-field simulations. The transition between different topological states could be realized via the assistance of external electric field imposed by PFM scanning probes. This work not only enriches the family of observed topological structures in ferroelectrics, but also provides a unique and effective way of designing intriguing topological structures through boundary condition engineering.

## Results

As one of the most extensively studied multiferroic materials, BFO is known to exhibit rich functionalities and various domain structures[35–37]. In particular, the topological states including the center-type domains[29,38], vortices[22,39] and bimerons[40] have been reported in confined BFO systems. To explore the promising possibility of polar topological states, 14.5 nm thick BFO films were grown on [001]-oriented (LaAlO$_3$)$_{0.29}$(SrTa$_{1/2}$Al$_{1/2}$O$_3$)$_{0.71}$ (LSAT) substrates, displaying the as-grown high-density self-assembly nanoislands (Fig. 1a). The averaged lateral size of the nanoislands is about 350 nm (Fig. 1a). The epitaxial nature of the BFO film is verified by the X-ray diffraction (XRD) scan (Fig. 1b) and the reciprocal space map (RSM) result (Fig. 1c). As revealed in Fig. 1c, the coexistence of rhombohedral BFO (R-BFO) and tetragonal-like BFO (T-like BFO) is stabilized in the film under the large compressive strain of 2.4% imposed by LSAT substrate. The domain structures of the BFO nanoislands are characterized by vector mode of PFM, which allows the simultaneous mapping of the vertical and lateral signals, including the amplitude and phase. As shown in Fig. 1d–f, the vertical PFM amplitude (V-amp.) and phase (V-pha.) images display as the doughnut-like OOP contrast for one nanoisland. The nanoisland constitutes of two parts: the yellow-colored core and the brown-colored periphery. Around the nanoislands, the flat BFO film share the same OOP phase contrast with the cores of nanoislands, which is further confirmed by the height and vertical PFM phase spacing profiles in Fig. 1h, i for one nanoisland (Fig. 1g). The result in Fig. 1i

suggests the alternative OOP polarization distribution in the manner of downward-upward-downward from flat BFO film, nanoisland periphery to nanoisland core.

To further resolve the 3D polarization distribution, a series of vector PFM mapping was conducted according to the method proposed previously[38,41] to determine the IP polarization distribution in the BFO nanostructures. The detailed reconstruction procedure is demonstrated in Fig. 2, with more details shown in Supplementary Fig. 1. The OOP polarization distribution for one nanoisland is revealed in Fig. 2a, showing the doughnut-like contrast. Then, by rotating the sample clockwise for 0°, 45° and 90° relating to the cantilever of PFM tips, the lateral PFM phases of the nanoislands at different angles can be found in Fig. 2b–d, displaying as the half-dark and half-bright contrast. Thus, the IP polarization distribution of one nanoisland is constructed as the unique domain structure of center-divergent or center-convergent arrangement. The special polarization distribution is prevalent in the high-density nanoisland array in the BFO film, with the detailed PFM analyses being displayed in Supplementary Fig. 1. Furthermore, the polarization distribution of one nanoisland is also confirmed by the high-angle annular dark-field imaging under the scanning transmission electron microscopy mode (HAADF-STEM) using Cs-STEM. For the IP direction, the polarization of one nanoisland displays as the four-quadrant contrast, which is further determined to be the center-divergent state, as shown in the low-magnification HAADF-STEM images (Fig. 2e, f) and atomic-resolved HAADF-STEM images (Fig. 2g–j). For the OOP direction, the polarization is in the direction of downward at the core of nanoisland (Supplementary Fig. 2-3). Thus, the polarization of one nanoisland is in the distribution of downward-upward from the core to the periphery of the nanoisland (Supplementary Fig. 4), with the polarization being continuous rotation. The OOP polarization for the surrounding matrix region (SMR) around the nanoisland is further determined as the downward direction (Supplementary Fig. 5). By combining the doughnut-like OOP polarization distribution (top panel in Fig. 2k) and four-quadrant IP polarization distribution (bottom panel in Fig. 2k and Supplementary Fig. 6), the three-dimensional polarization distribution for the combined area (CA) consisting of one nanoisland and SMR could be reconstructed, as schematized in Fig. 2l–n, which is reminiscent of the skyrmionium to some extent[13]. However, considering the fact that the polarization changes from core region to SMR region in the BFO nanostructures are not strictly calculated as $2\pi$, the polarization state is thereby defined as the ferroelectric two-order radial vortex, displaying as two nested concentric vortices. Meanwhile, the ferroelectric two-order radial vortex is expected to enrich the polar topological states and exotic physical phenomena in ferroelectric materials. The spontaneous occurrence of the polar two-order radial vortex in the as-grown BFO nanostructures implies that it is the favorable state stabilized by boundary condition engineering, which is observed prevalently in BFO film. The formation of this topological state could be attributed to the combined contributions of depolarization field, strain relaxation and charge accumulation. On the one hand, the BFO films fabricated via high deposition flux tend to form the nanoislands[38,41]. At the core of the nanoislands, the elemental non-stoichiometry of Bi and Fe in Supplementary Fig. 7 is expected to from the atomic interdiffusion between the film and the substrate during the film deposition and the annealing procedure due to the spontaneously formed dipole disclinations in the cores of the nanoislands[41–44]. Thus, the charge is accumulated due to the elemental difference (Supplementary Fig. 7), oxygen vacancies or other potential charged carriers[45,46], thereby stabilizing the tail-to-tail charged domain walls and forming the center-divergent polarization along in-plane direction (Fig. 2e–j). On the other hand, the coupling effects between the depolarization field and the lattice mismatch strain at heterointerface in the BFO films stabilize the alternative out-of-plane polarization distribution, with the nanoisland cores as the preferred nucleation sites of ferroelastic domains. As a

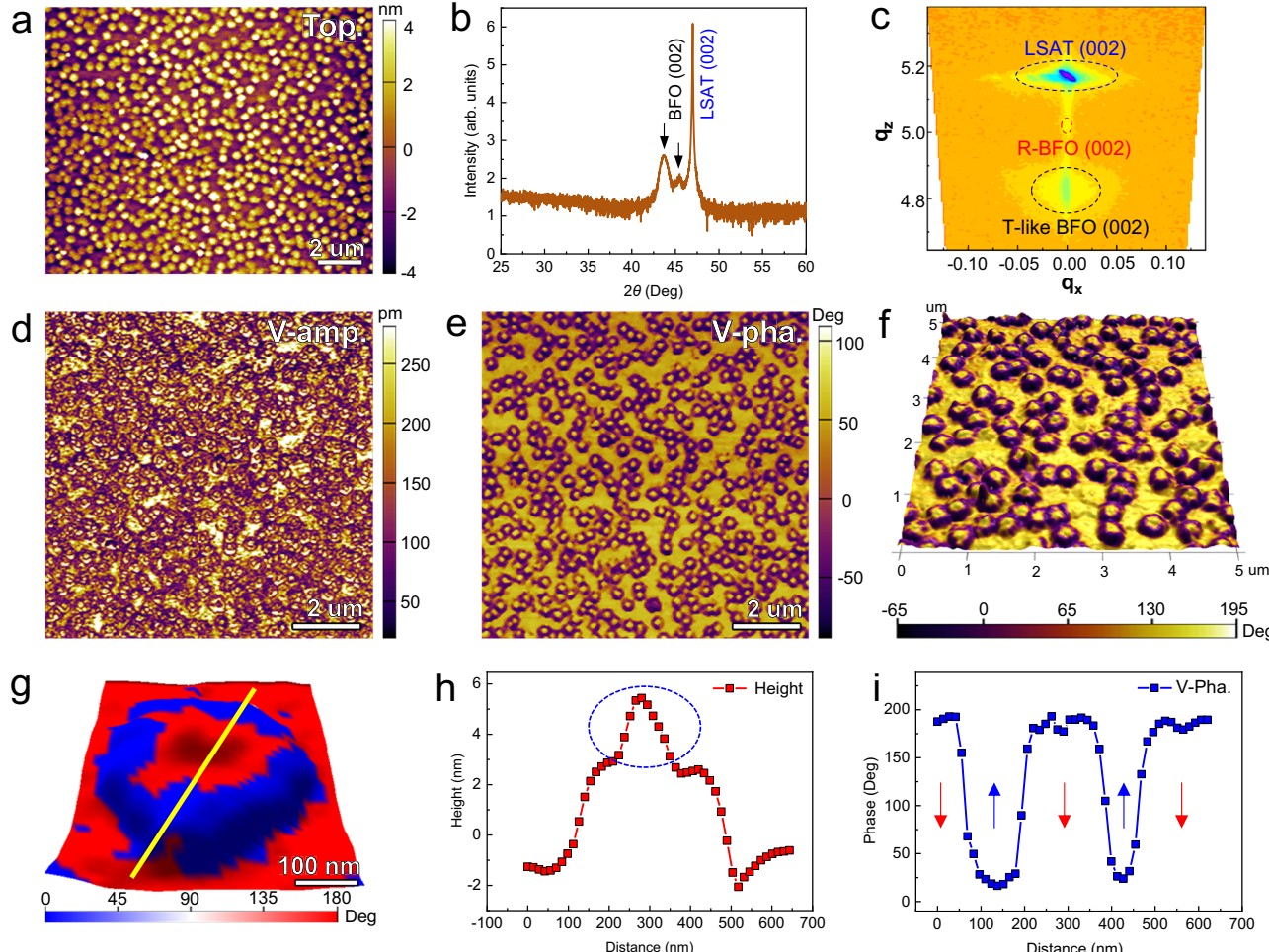

**Fig. 1 | PFM analyses of high-density BFO nanoislands. a** Topography of BFO (001) thin film with self-assembled nanoislands. **b** X-ray diffraction results showing the pseudocubic 002 peaks of the films. **c** Reciprocal space map recorded around the LSAT 002 Bragg peaks. **d, e** Vertical PFM amplitude (V-amp.) and vertical PFM phase (V-pha.) images of the domain patterns in the BFO film. **f** Enlarged OOP phase image superimposed with the topography image. **g** OOP phase image of one BFO nanoisland. **h, i** Height and vertical PFM phase spacing profiles along the yellow line in (**g**). The hump denoted by blue ellipse in (**h**) is derived from the additional adsorbates from air.

result, the two-order radial vortex with alternative out-of-plane polarization and center-divergent in-plane polarization is obtained in the BFO films.

Furthermore, the polarization configurations of the nanostructures can be modulated by the size of the nanostructures (defined in Supplementary Fig. 8), thereby generating multiple polar topological states, as summarized in Supplementary Table 1. In Fig. 3 and Supplementary Fig. 9-11, three kinds of polarization patterns are revealed, with the averaged sizes of nanoislands changing from 100 nm, 200 nm to 400 nm. For the case of 100 nm nanoislands, the morphology, vertical amplitude and phase images are displayed in Supplementary Fig. 9 and Fig. 3a, b. The uniform phase contrast is observed inside the nanoislands, which is surrounded by the different phase contrast of the SMR. Figure 3c is the phase spacing profile for one nanostructure (inset in Fig. 3b), suggesting the nearly 180° out-of-plane phase difference between two regions. The arrows in Fig. 3c suggest the opposite OOP polarization directions. The IP polarization distribution for the nanoisland displays the similar center-divergent pattern as the case of 350 nm nanoisland. By combining the IP and OOP polarization distribution, the 3D polarization pattern of the CA is constructed as one-order radial vortex (Fig. 3d), with the change of azimuth angle being 0.5π (Supplementary Fig. 12a). The CA includes the single nanoisland (inside the solid circle) and SMR (region between

solid circle and dotted circle) shown in the inset in Fig. 3d. Increasing the size of nanoisland to 200 nm (see the morphology image in Supplementary Fig. 10), the vertical amplitude and phase images of the nanoislands and the SMRs display the different contrast, as shown in Fig. 3e, f. The similar doughnut-like OOP polarization distribution is revealed inside the nanoislands (Fig. 3f), but with the SMR sharing the same vertical phase contrast as the nanoisland periphery, which is further confirmed by the phase spacing profile in Fig. 3g. As a result, the polarization pattern for the CA including the 200 nm nanoisland is also reconstructed and defined as the one-order radial vortex, also with the change of azimuth angle being 0.5π (Supplementary Fig. 12b). When the averaged size of the nanoisland is 400 nm (see the morphology image in Supplementary Fig. 11), the vertical phase contrast nearly displays as the doughnut-like OOP polarization distribution reminiscent of the two-order radial vortex in Fig. 1e, excluding the emergence of the localized opposite OOP phase in the middle of nanoisland core (Fig. 3i, j). As shown in Fig. 3k, l, the phase change of the OOP polarization from the center of the nanoisland core to the SMR is calculated as 2.5π (Supplementary Fig. 12d). Thereby, the constructed polarization pattern in Fig. 3l is defined as the three-order radial vortex. In a word, by changing the size of nanostructures, three kinds of polar topological states are stabilized, including the one-order radial vortex, two-order radial vortex and three-order radial vortex. It is

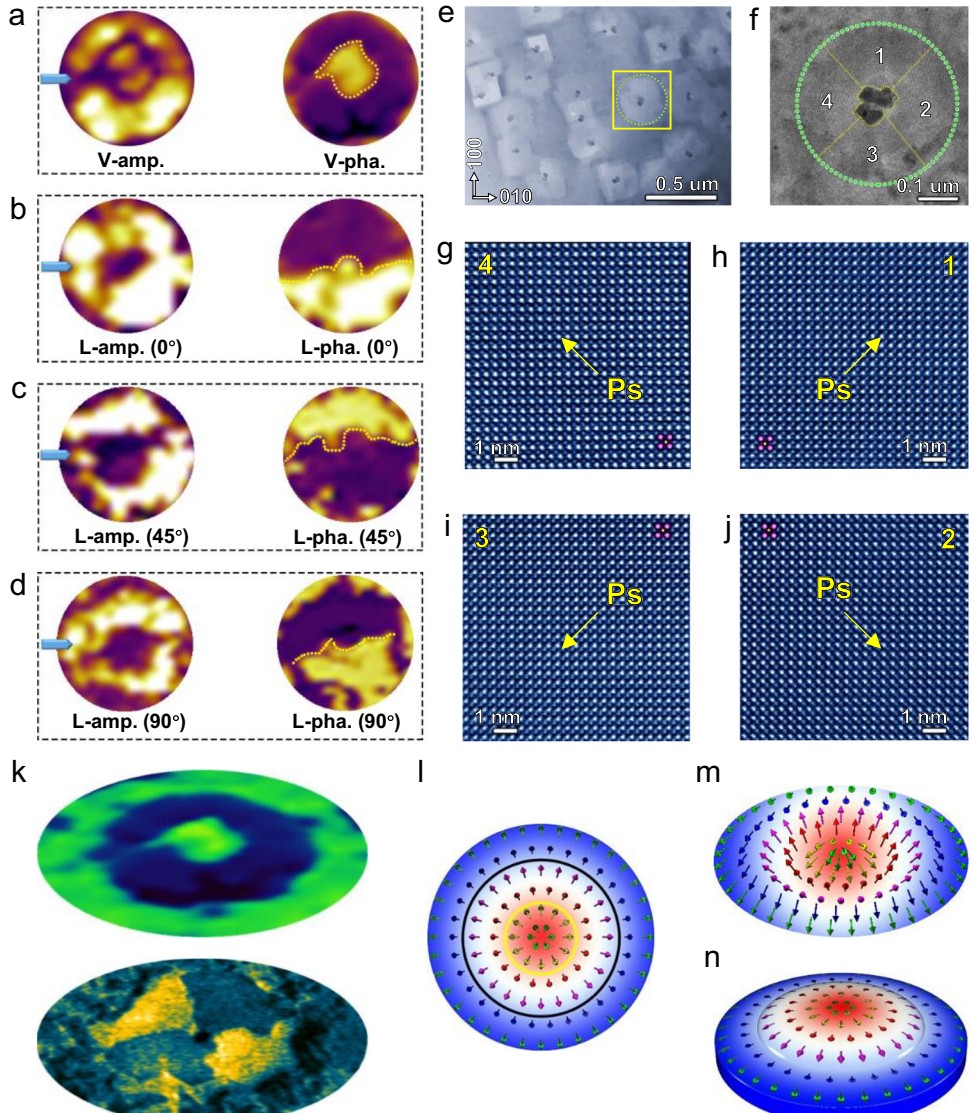

**Fig. 2 | Reconstructed polar two-order radial vortex in one nanostructure.**
**a** The vertical PFM amplitude (V-amp.) and vertical PFM phase (V-pha.) images of one nanoisland. **b–d** The lateral PFM amplitude (L-amp.) and lateral PFM phase (L-pha.) images of one nanoisland with sample rotation for 0° (**b**), 45° (**c**) and 90° (**d**), respectively. Blue arrows in (**a–d**) denoting the cantilevers of PFM tips. **e** A planar-view HAADF-STEM image showing the nanoislands in BFO film. **f** Enlarged HAADF-

STEM image for one nanoisland. **g–j** Polarization distribution of BFO corresponding to the regions numbered 1, 2, 3 and 4 in (**f**), respectively. **k** Enlarged V-pha. image (top panel) and HAADF-STEM image (bottom panel) around one nanoisland, displaying the out-of-plane and in-plane polarization distribution, respectively. **l–n** Reconstructed 3D polarization distribution around one nanoisland, suggesting the polar topological state of two-order radial vortex.

worthwhile noting that the nanoislands would merge with each other and form the flat surface in thicker BFO films (about 20 nm), thereby stabilizing the labyrinthine domains, as shown in Supplementary Fig. 13.

The evolution of domain pattern and their topological features with diameter of the BFO nanoislands are further investigated by the phase-field simulations (details in Methods). The schematic of the phase-field model for the disc-shaped BFO nanostructures is shown in Supplementary Fig. 14. To mimic the experimental conditions, three different sizes of the disc-shaped BFO nanostructures with diameters of $d$, $2d$, and $3d$ are simulated and compared, as shown in Fig. 4a–c. It should be admitted that the theoretical transition size is slightly smaller than the experimental size, which can be attributed to a higher theoretical depolarization field where the perfect charge screening is assumed. However, it can be clearly seen that the trend for the size dependent transition agrees remarkably well. With a priori setting of the 180° circular domains and a subsequent annealing treatment

analogous to the experimental preparation, different topological structures were formed in the three BFO nanostructures (Fig. 4d–f). Single doughnut-like circular domains can be stabilized in the BFO nanostructure when the diameter is $d$, similar to the bubble-like structure observed in the PTO/STO system[24]. The planar distribution of OOP polarization and surface integration of the Pontryagin density reveals that this circular domain structure is a polar one-order radial vortex (Fig. 4d). As the nanostructure diameter increases from $d$ to $2d$, the double nested doughnut-like domain could also be formed in the BFO nanostructure, confirming the formation of polar two-order radial vortex (Fig. 4e). Notably, when the nanostructure diameter further increased to $3d$, a triple nested circular domain pattern occurs in the BFO nanostructure (Fig. 4f). In this case, such structure is determined to be a three-order radial vortex. The local OOP polarization distribution of the BFO nanostructures with the three sizes are plotted (Fig. 4g–i), showing the stabilization of three different topological phase features, i.e., one-order radial vortex, two-order radial vortex,

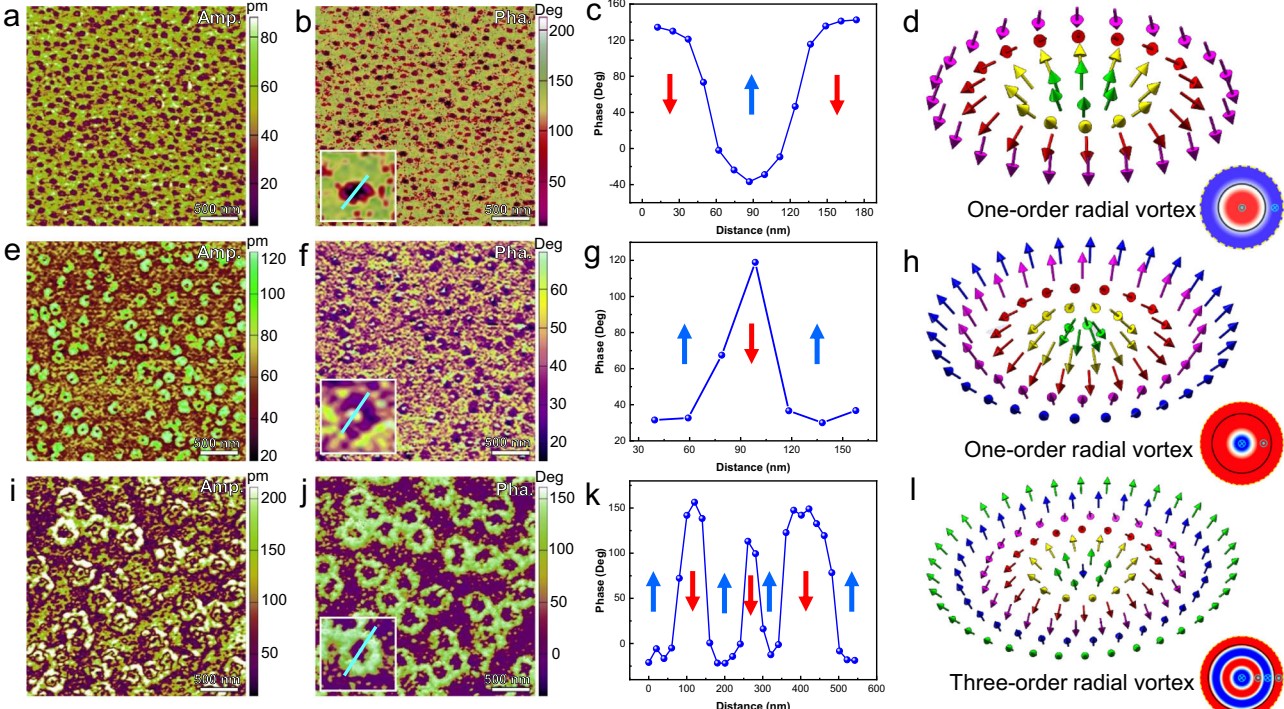

**Fig. 3 | Topological transition as a function of the sizes of nanoislands. a–b** V-amp. and V-pha. images of the BFO film with the averaged size of nanoislands being 100 nm. **c** Phase spacing profile around one nanoisland in the inset of (**b**). **d** Reconstructed 3D polarization pattern of one-order radial vortex. The schematic at the bottom right corner showing the OOP polarization distribution around one nanoisland. **e, f** V-amp. and V-pha. images of the BFO film with the averaged size of nanoislands being 200 nm. **g** Phase spacing profile around one nanoisland in the

inset of (**f**). **h** Reconstructed 3D polarization pattern of one-order radial vortex. The schematic at the bottom right corner showing the OOP polarization distribution around one nanoisland. **i, j** V-amp. and V-pha. images of the BFO film with the averaged size of nanoislands being 400 nm. **k** Phase spacing profile around one nanoisland in the inset of (**j**), **l** Reconstructed 3D polarization pattern of three-order radial vortex. The schematic at the bottom right corner showing the OOP polarization distribution around one nanoisland.

and three-order radial vortex by varying the size of the nanostructures, which agrees qualitatively well with the experimental observations.

To further gain physical insights into the creation mechanism for these multi-order radial vortices, different energy contributions within the one-order, two-order and three-order radial vortices in BFO nanostructures were investigated from phase-field simulation. For simplicity, the BFO nanodisk diameter was fixed at 3 *d* to analyze the individual energy density differences among pre-designed one-order, two-order and three-order radial vortices. As shown in Supplementary Fig. 15, increasing the vortex order from one-order to three-order introduces more domain walls, raising the gradient energy. However, this is effectively compensated by reductions in Landau, electrostatic, and elastic energies, which relieve system strain and efficiently screen the depolarization field, allowing the three-order radial vortex to stabilize. Therefore, from a theoretical perspective, the Landau energy, elastic energy, and electrostatic energy could serve as the driving forces for multi-order radial vortex formation in BFO nanostructures.

To illuminate the polarization switching behaviors of the polar two-order radial vortex in Fig. 2, localized PFM phase-field hysteresis loops and amplitude-field butterfly loops are compared in Supplementary Fig. 16 for three representative regions, including the nanoisland core (numbered region 1), nanoisland periphery (numbered region 2) and SMR (numbered region 3). The localized coercive fields for three regions could be obtained from PFM phase-field loops and PFM amplitude-field loops in Supplementary Fig. 16b-d. The coercive field of nanoisland core in Supplementary Fig. 16b is calculated as 4500 kV/cm, which is larger than that of nanoisland periphery (2264 kV/cm in Supplementary Fig. 16c) and SMR (3500 kV/cm in Supplementary Fig. 16d), suggesting the tougher possibility of polarization switching at the same applied field. Then the topological

transitions in BFO films are further discussed under the stimulation of external electric fields. The vertical PFM phase images of the initial state and poled state by the *dc* bias voltages of +40 V and -40V are displayed in Fig. 5a–c, indicating that the changed OOP polarization distribution for the nanostructures including the nanoislands and the SMR. To elucidate the detailed topological transitions modulated by *dc* bias electric field, three kinds of topological transitions are highlighted by red (Type 1), green (Type 2) and yellow (Type 3) circles in Fig. 5a–c. The changes of OOP polarization distribution and topological transitions are further schematized in Fig. 5d–i. For the Type 1 in Fig. 5d, e and Supplementary Fig. 17a, the initial domain state of one nanostructure is three-order radial vortex. After the electrical writing experiment using the voltage bias of 40 V, the three-order radial vortex transforms into the non-topological domain (abbreviated as NTD). Then after additional stimulation of -40 V at the same region, the domain state of the nanostructure is stabilized as two-order radial vortex. During this process, the topological charge changes from $|Q| = 0.707$ to $|Q| = 0$ and finally $|Q| = 0$ (Supplementary Fig. 18a). For the Type 2 in Fig. 5f, g and Supplementary Fig. 17b, the topological transition process modulated by opposite voltage bias is determined to be from two-order radial vortex to NTD and finally switching back to two-order radial vortex, accompanying with topological charge remaining constant ($|Q| = 0$) (Supplementary Fig. 18b). For the Type 3 in Fig. 5h, i and Supplementary Fig. 17c, the transition from two-order radial vortex to NTD and finally one-order radial vortex is schematized, with the topological charge changing from $|Q| = 0$ to $|Q| = 0$ and finally $|Q| = 0.707$, as shown in Supplementary Fig. 18c. As a result, three kinds of topological transitions could be concluded under the stimulation of opposite voltage bias. The concomitant changes of topological charges (Supplementary Fig. 18) suggest the feasibility of employing the

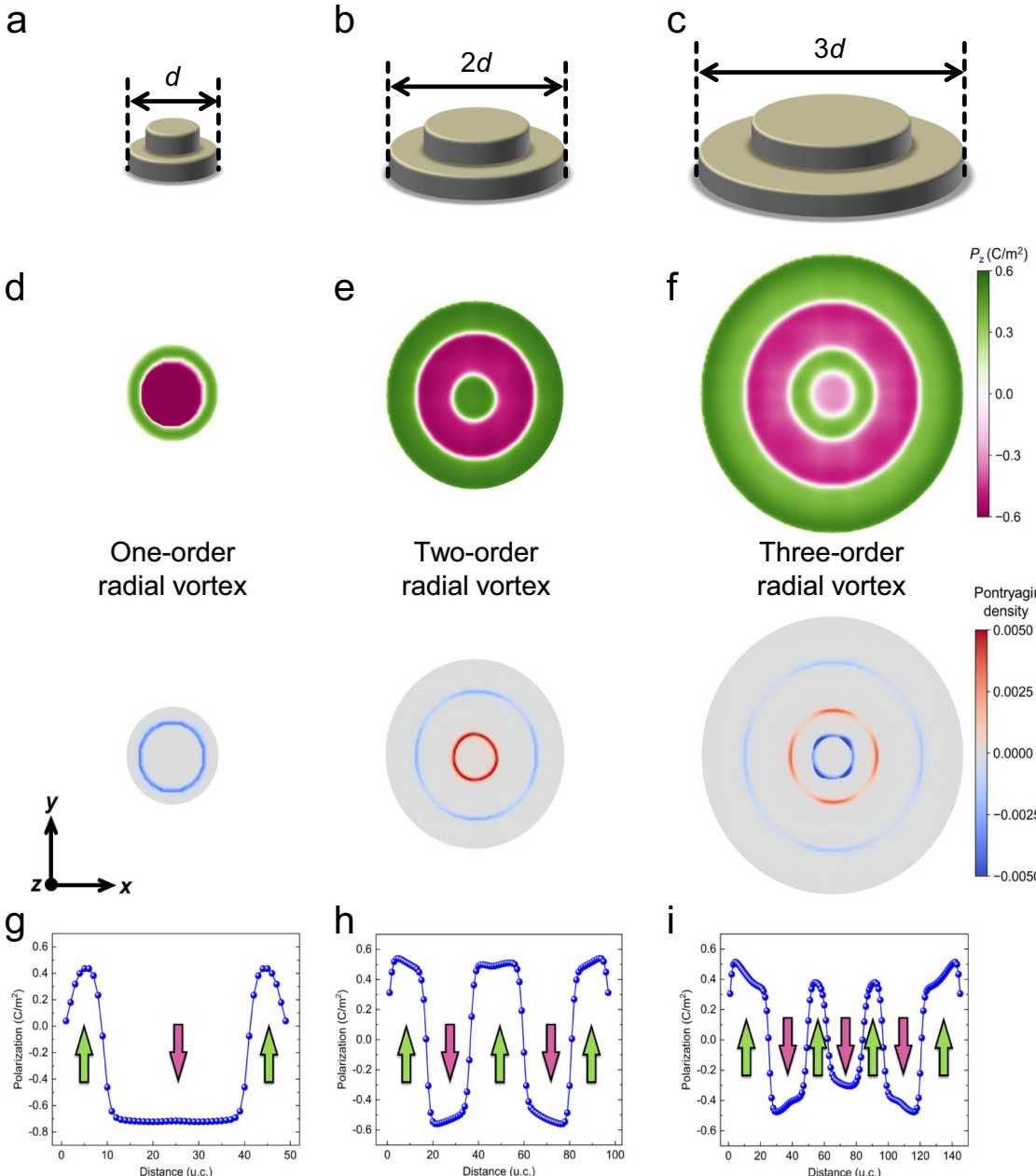

**Fig. 4 | Phase-field simulations of polar topological structures of BFO islands with different diameters. a–c** Schematics of the nanostructures including the disc-shaped BFO islands and the surrounding BFO matrix, with the diameters of the nanostructures being $d$, $2d$, and $3d$, respectively. **d–f** Planar view of out-of-plane polarization and Pontryagin density distributions in BFO nanostructures with diameters of $d$, $2d$, and $3d$. **g–i** The local out-of-plane polarization distributions along a horizontal line through the center of BFO nanostructures with diameters of $d$, $2d$, and $3d$.

topological charges as the digital bits in the promising vortex-based multistate nonvolatile memory devices[10].

In summary, we report the observation of self-assembled polar two-order radial vortex and other multi-order radial vortices in high-density $BiFeO_3$ nanostructures through boundary condition engineering. The polar two-order radial vortex, exhibits a doughnut-like pattern of out-of-plane polarization and four-quadrant in-plane polarization. Furthermore, by combining the phase-field simulations, it is confirmed that the topological states could be tuned by varying the size of the BFO nanostructures. Multi-order radial vortices with adjustable topological charges, such as the one-order radial vortex and three-order radial vortex, have been stabilized in nanostructures of different sizes. Transitions between these topological states can also be achieved under the application of an external electric field. The

discovery of polar two-order radial vortex and the transition between various multi-order radial vortices highlight the rich diversity of topological structures and offer a promising strategy for multi-state, non-volatile ferroelectric memory devices.

## Methods

### Film deposition details

Using pulsed laser deposition (PLD) with a Coherent ComPex PRO 201 F KrF ($\lambda = 248$ nm) excimer laser, a series of epitaxial BFO thin films on LSAT (001) substrates were deposited. The LSAT (001) substrates used here are commercial substrates without extra chemical or heat treatment. Before deposition, the substrates were dipped in the 90% alcoholic solution for 12 hours to clean the organic pollutant and dusts. The substrates were affixed in the substrate plate using the silver paint

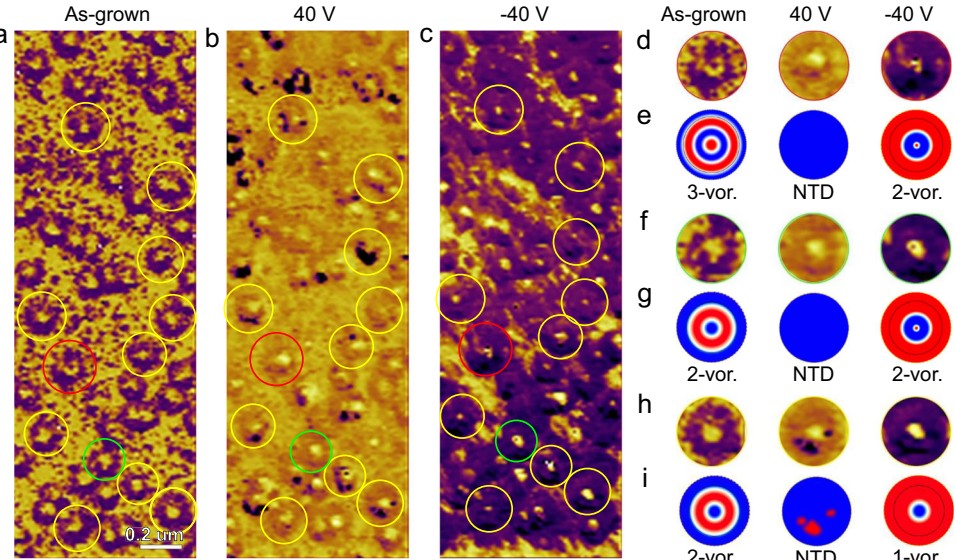

**Fig. 5 | Changes of polarization patterns under external electric fields. a** Vertical PFM phase image of the initial domain pattern. **b** Vertical PFM phase image after writing experiment with a voltage of 40 V. **c** Vertical PFM phase image after additional writing experiment with opposite voltage of −40 V. The red, green and yellow circles denote three kinds of topological transitions. **d** Enlarged vertical PFM phase images of one nanostructure highlighting by red circle in (**a–c**). **e** Corresponding schematics of (**d**), suggesting the transition from three-order radial vortex to NTD and finally radial vortex. **f** Enlarged vertical PFM phase images of one nanostructure highlighting by green circle in (**a–c**). **g** Corresponding schematics of (**f**), suggesting the transition from two-order radial vortex to NTD and finally two-order radial vortex. **h** Enlarged vertical PFM phase images of one nanostructure highlighting by yellow circle in (**a–c**). **i** Corresponding schematics of (**h**), suggesting the transition from two-order radial vortex to NTD and finally one-order radial vortex. The one-order, two-order and three-order radial vortices are abbreviated to 1-vor., 2-vor. and 3-vor., respectively.

solution and transferred into the main chamber of the PLD equipment. Then the substrates were heated to 850 °C for 20 minutes to clean the substrate surfaces and then cooled slowly down to the film deposition temperature at a rate of 5 °C min⁻¹. Before growing the BFO layers, the velocity for the substrate rotation motor and the target DC rotation motor were set to be 47 deg s⁻¹ and 71 deg s⁻¹, respectively. At the same time, the raster motor mode was selected for the target carousel motor, with the maximum speed being 30 deg s⁻¹ and minimum speed being 1 deg s⁻¹. The deposition of BFO films used the 1 mol% Bi-enriched BFO target, which was pre-sputtered for 20 minutes at 850 °C to clean the surface with the shutter being closed. When growing the BFO layers, the shutter was opened, a repetition rate of 8 Hz, substrate temperature of 800 °C, oxygen partial pressure of 12 Pa and laser energy of 2 J cm⁻² were used. The distance between target and substrate was set to be 60 cm. Under the above deposition condition, the height of the laser plume was observed to be about the 1/2 of the distance between target and substrate. After deposition, these films were annealed at 800 °C in an oxygen partial pressure of 266 Pa for 20 minutes and then cooled slowly to room temperature at a rate of 5 °C min⁻¹.

## PFM observations

The PFM characterization was performed using an Asylum Research Cypher S atomic force microscope (Oxford Instruments) at room temperature. Both vertical and lateral PFM images were simultaneously acquired through Vector PFM mode, with data validity cross-verified by Dual AC Resonance Tracking (DART) methodology. Ti/Ir (5/20)-coated conductive probes (ASYELEC-01-R, spring constant: 2.8 N/m) were employed, with contact resonance frequencies optimized at 350 kHz (vertical) and 760 kHz (lateral) through thermal noise calibration. Prior to local hysteresis measurements, cantilever sensitivity was rigorously calibrated via the GetReal method[47]. Domain switching dynamics were investigated by applying DC bias voltages in DART mode, while nanoscale domain lithography was executed through sequential voltage patterning using the AFM lithography mode.

## TEM sample preparation

STEM specimens were prepared through a standardized mechanical processing process: bulk samples were sectioned, epoxy-bonded, mechanically thinned to <20 μm thickness, dimpled to <5 μm center thickness, and precision-finished via Ar⁺ ion milling (Gatan 691 PIPS). To minimize beam-induced damages, a multi-stage ion milling strategy was implemented: initial coarse milling at 7° incidence angle and 4.5 keV with liquid nitrogen cooling, followed by a final low-energy polishing step (0.1 keV, 10 min, ±5° beam oscillation).

## STEM observation

HAADF-STEM characterization was conducted using a double Cs-corrected ThermoFisher Spectra 300 (scanning) transmission electron microscope (CEOS probe/imaging correctors) operated at 300 kV. The probe semi-convergence angle was optimized to 25 mrad, with HAADF detector inner/outer collection angles set at 71/200 mrad to ensure optimal Z-contrast sensitivity.

## STEM result analyses

The drift-corrected frame integration function was used for frame series in Velox software to create a single image, aiming to optimize contrast while minimizing beam-induced specimen drift artifacts. To enhance signal-to-noise ratio, raw HAADF-STEM images underwent Wiener deconvolution coupled with a low-pass frequency filter (cutoff at the instrument's theoretical resolution limit). Atomic column positions were quantitatively determined through sub-Å precision 2D Gaussian fitting implemented in MATLAB[48], enabling systematic mapping of B-site cation displacements.

## X-ray diffraction and reciprocal space mapping

Crystallographic analysis was conducted using a high-resolution Bruker D8 Advance X-ray diffractometer (Cu Kα radiation source, λ = 1.5406 Å), employing $\theta$-$2\theta$ scans and reciprocal space mapping to resolve the out-of-plane lattice parameters and phase structures.

## Phase-field simulations

In the phase-field approach, the evolution of the order parameters spontaneous polarization vector (**P**) and oxygen octahedral tilt (**θ**) in the islands on the surface of BFO films grown on LSAT substrates is governed by the time-dependent Ginzburg-Landau equation:

$$\frac{\partial \phi}{\partial t} = -M \frac{\delta F}{\delta \phi} \qquad (1)$$

where **ϕ**, $t$ and $M$ denote the order parameter (either **P** or **θ**), the evolution time step and the dynamic coefficient, respectively. The total free energy $F$ of the BFO island has the contributions from the individual energy densities, i.e., the Landau/chemical, elastic, electrostatic, and polar/rotation gradient energy densities:

$$F = \int \left( f_{Landau} + f_{elastic} + f_{electric} + f_{gradient} \right) dV \qquad (2)$$

Detailed expressions of these energy densities, materials parameters as well as the numerical simulation procedure are described previous reports[49–51].

The discrete grid points of $200\Delta x \times 200\Delta y \times 90\Delta z$ with a grid spacing of 0.4 nm are used to describe the BFO system consisting of a disc-shaped nanoisland surrounded by matrix region, substrate and air layer (see Supplementary Fig. 14). Corresponding to the experimental results, three different sizes of BFO islands are considered and their diameters are set to the $d$, $2d$, and $3d$, with $d$ being 48 grids, respectively. A thin STO dielectric layer is also introduced at the top and bottom of the BFO island to simulate the depolarization field effect at the interface. The total height of BFO island is 36 grids inside the simulation mesh. Periodic boundary conditions are assumed in the two in-plane dimensions of the entire model, while a superposition method is applied in the thickness dimension[52]. To simulate the evolution of polar radial vortices with disk diameter in BFO nanoislands, the initial set-up for the simulation is a series of the 180° circular domains with a small random noise (<0.01 μC/cm²). Then they relaxed to the stable states through an annealing process.

## Reporting summary

Further information on research design is available in the Nature Portfolio Reporting Summary linked to this article.

## Data availability

The data that support the findings of this study are provided in the article and the Supplementary Information. The data sets generated and analyzed during the current study are available from the corresponding author on reasonable request.

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

## Acknowledgements

This work is supported by National Natural Science Foundation of China (No. 52201018 (W.R.G.), No. 51971223 (Y.L.Z.), No. 51922100 (Y.L.T.), No. 92166104 (Z.H.), No. U21A2067 (Y.W.)), Guangdong Basic and Applied Basic Research Foundation (2021A1515110291 (W.R.G.), 2023A1515012796 (W.R.G.)), Guangdong Provincial Quantum Science Strategic Initiative (GDZX2202001, GDZX2302001, GDZX2402001 (X.L.M.)), the Open Fund of the Microscopy Science and Technology, Songshan Lake Science City (202401202 (W.R.G.)), the Key Research Program of Frontier Sciences CAS (QYZDJ-SSW-JSC010 (X.L.M.)) and Shenyang National Laboratory for Materials Science (L2019R06 (X.L.M.), L2019R08 (Y.L.Z.), L2019F01 (Y.L.T.), L2019F13 (Y.J.W.)), Y.L.T. acknowledges the Scientific Instrument Developing Project of CAS (YJKYYQ20200066), and the Youth Innovation Promotion Association of CAS (Y202048). Z. H. acknowledges a financial support by the Fundamental Research Funds for the Central Universities (2023QZJH13). Y.J.W. acknowledges the Youth Innovation Promotion Association CAS (No. 2021187). The phase-field simulation was performed on the MoFang III cluster on Shanghai Supercomputing Center (SSC). X.G. is supported by the National Natural Science Foundation of China (No. 52202151) and China Postdoctoral Science Foundation (No. 2022M722715).

## Author contributions

W.R.G. and X.G. contributed equally to this work. X.L.M. conceived the project on the architecture of quantum materials modulated by ferroelectric polarizations; W.R.G., Y.L.Z. and X.L.M. designed the sample structures and subsequent experiments. W.R.G. performed the thin-film growth and PFM observations. W.R.G., Y.L.T. performed the TEM and STEM observations. X.G., D.M., Y.W. and Z.H. performed the phase-field simulations. D.M. and Y.J.W. carried out digital analysis of the STEM data. All authors participated in discussion and interpretation of the data.

## Competing interests

The authors declare no competing interests.
