## [Transparent Peer Review file · Nature Communications]

Observation of multi-order polar radial vortices and their topological transition

Corresponding Author: Professor Xiu-Liang Ma

Version 0:

Reviewer comments:

Reviewer #1

(Remarks to the Author)

This work done by W. R. Geng et al. reported the observation of polar skyrmionium, a new polarization state, in BFO thin film prepared by PLD. Detailed PFM and TEM results unveil the existence of the $n\pi$ -skyrmion in islands with different sizes, and the transition between different $n\pi$ -skyrmion under external electric field. Overall, the work contributes an interesting and important discovery. Therefore, I recommend the publication of this manuscript after addressing the following questions.

1. One of the core findings of this study is the modulation of $n\pi$ -skyrmions by tuning the size of the islands. Is the corresponding film thickness the same? Could you provide more details on the film growth methods used? It is strongly recommended to include corresponding morphology images, as morphology significantly affects PFM phase and amplitude from the perspective of PFM testing.
2. Extended Data Fig. 7 shows the elemental distribution in one nanoisland. From the figure, the central region of each nanoisland (indicated by the white dashed circle) shows significantly different elemental composition, with a lack of Bi and Fe, while Al and Sr are enriched. The former originates from the film and the latter from the substrate. Combining with the results from Fig. 1h, the central position is clearly protruded (the height of this island also differs significantly from the TEM results in Extended Data Fig. 2). This seems a bit strange. Are there nanoislands on the substrate? Or could it be that diffusion occurred between the film and the substrate? Or are there defects at the center of the nanoislands? The TEM results in the article do not show obvious defects.
3. The authors achieved the transformation from skyrmion to NTD using an applied electric field. Can the reversible transformation between $n\pi$ -skyrmions be further investigated experimentally? After all, this is essential for truly realizing a "promising candidate of information carriers."
4. Lines 47-49, references 11 and 12 do not appear to be correct.
5. The method of calculating the coercive field using amplitude-voltage butterfly loops in Extended Data Fig. 9 seems unscientific, and voltage should not be used as a substitute.

Reviewer #2

(Remarks to the Author)

The polar topological states, especially the skyrmion family, exhibit novel functionalities distinct from the bulk domains. Despite the remarkable progress in the experimental observations and theoretical study on polar skyrmion, the skyrmionium and deterministic control of target skyrmions has not been experimentally observed to date. In this manuscript, the authors report the observation of self-assembly polar skyrmionium and other target skyrmions in high-density BiFeO₃ nanostructures through boundary condition engineering. Firstly, the polar skyrmionium with the topological number of zero, also called as 2π -skyrmion, was observed which is characterized as the doughnut-like pattern of out-of-plane polarizations. Secondly, it was found that multiple target skyrmions with adjustable topological numbers can be stabilized in BFO nanostructures with

different sizes, including the π -skyrmion and 3π -skyrmion. Finally, they found that topological transitions between different topological states can be achieved via the stimuli of external electric field. These results enrich the family of topological structures in ferroelectrics. Overall, the current work is interesting, but I believe that the current version is below the level of Nature Communications. I cannot recommend publishing it until all my doubts are satisfactorily answered.

(1) The magnetic skyrmion Hall effect mentioned several times is a classical dynamic effect, but the skyrmionium in this article is actually a topological structure bound in high-density self-assembled nanoislands. Suggest modifying the relevant statements.

(2) The manuscript focuses on the study of $n\pi$ -skyrmion, which involves the key concept of topological charge (skyrmion number) Q . Therefore, Q needs to be clearly defined. The Q value of $n\pi$ -skyrmion should be demonstrated in both experiments and calculations, rather than being roughly assumed. Please refer to previous work (Nature communications, 2015, 6(1): 8542).

(3) In Figure 3g, the phase difference is only 0.5π instead of π , so its Q cannot be 1. In fact, for cases where the phase difference between the core and peripheral polarization is less than $n\pi$, Q is not an integer n . The data analysis in the manuscript should be more rigorous.

(4) To my knowledge, for skyrmions in tetragonal PTO or PZT, polarization rotation π from the core (upward) to the periphery (downward) is necessary to ensure that Q is 1. However, for BFO nanodots under misfit strain, the core and peripheral polarizations in the skyrmion-like structure are difficult to point to out-of-plane distributions, resulting in Q being a non-integer. Therefore, in the manuscript, perhaps $Q=0$ for the " 2π -skyrmion", but for the " π -skyrmion" mentioned in the article, I believe the Q value of " π -skyrmion" is not 1 (maybe less than 1). In my opinion, it is currently impossible to prove that the structures in the manuscript are members of the skyrmion family, perhaps they can be called quasi-skyrmion or skyrmion-like structures.

(5) Some typos, for example, "an unique" in Line 86 should be changed to "a unique"; Some capitalization issues, for example, "the uniform" in Line 160 should be changed to "The uniform"; Some inappropriate expressions, for example, "high performance" in Line 36 should be changed to "high-performance", the statement in line 201-202 should be further polished, etc. Please check the text and references carefully and thoroughly.

(6) In Figure 3, d, h, and l correspond to c, g, and k, respectively, right? If so, the polarization directions of d and l are drawn incorrectly. In addition, the polarization orientation of d and h does not correspond to the schematic at the bottom right corner as well.

(7) The statements in Line 202-205 should be carefully checked. Figure 4 discusses the influence of nanoisland diameter on topological domain structures, which actually is not related to topological phase transitions. In my opinion, the topological phase transition is the transition between states with different topological charges under the action of adjustable parameters (such as temperature, external electric field, etc.). However, the size of the specific nano-island is fixed and cannot be changed.

(8) The size of the nano-islands has a significant effect on the domain structures. In Figure 4, the simulation used a much smaller nano-island size than the experimental one, but obtained similar results. Is this reasonable as a theoretical verification?

(9) In the phase-field model, is the surrounding matrix region (SMR) around the nanoisland considered?

Reviewer #3

(Remarks to the Author)

The manuscript "Observation of polar skyrmionium and topological transition between $n\pi$ -skyrmions" by W. R. Geng et al. reports the observation of polar skyrmionium and family of target skyrmions with different winding numbers in multiferroic BiFeO₃. In the manuscript, the authors demonstrate how nanostructures in the deposited film induce and affect the self-assembly of skyrmionium and $n\pi$ -skyrmions. The authors also provide a method to manipulate topological numbers using an external electric field. Given that skyrmionium is expected to resolve the problem of vanishing skyrmions in the proposed skyrmion devices, the work seems timely, and the conclusions seem well-supported by the experiments. Here are some comments and revisions I would like to see addressed in the manuscript before publication.

1. In the introduction, the authors explain that one of the main motivations for studying the $Q=0$ skyrmions or skyrmionium is to avoid the skyrmion Hall effect, which is detrimental for the skyrmion racetrack memory-related applications. On the other hand, the skyrmionium in the BFO that the authors discovered seems to be an inherent property of the nanostructures. This makes skyrmioniums difficult to move and reduces their merits as lossless information carriers in racetrack-related devices. While the authors don't explicitly claim that the polar skyrmionium or $n\pi$ -skyrmions can be used in racetrack devices, I think this can still be considered a discrepancy to readers.

2. Related to the previous question. If the polar skyrmioniums are immobile, it would be great if the authors could elaborate on how to utilize polar skyrmioniums and target skyrmion family.

3. The authors explained that the film is under large compressive strain, and the rhombohedral and tetragonal phases coexist. Is there a connection between this and the formation of the nanoislands, or is the mixture of phases uniform?

4. It would be better if the creation mechanism for the skyrmionium is explained in more detail. While the authors reference theoretical works or the balancing mechanism between electrostatic energy and elemental non-stoichiometry, it is still difficult to see the full extent of components that determine the formation of the target skyrmions and their internal structures.

5. Also, regarding the formation mechanism, does the magnetic state of the synthesized BFO play any role? What is the magnetic phase of synthesized BFO, and do they affect the formation of proposed textures?

6. In the latter part of the manuscript, the authors use different bias voltages to control the topological number. While this suggests a possibility of controlling topological numbers, the application of voltage always seems to work in the direction where the number is decreased or maintained. Is increasing the topological number also possible, for instance, by applying reverse voltage or some additional process?

7. The vertical PFM phase image in Figure 5 suggests that a mixture of 2π and 3π skyrmions can be initialized. Is this because the size of the nanostructure in the same film is non-uniform, or does this also have to do with the metastability of the polar skyrmions? If such metastability is essential for determining topological numbers, analyzing the statistical distribution of skyrmions with different topological numbers might also provide insight into the energetics of skyrmion formation.

Version 1:

Reviewer comments:

Reviewer #1

(Remarks to the Author)

I find that the authors have well addressed the raised concerns. Therefore, I am happy to recommend the paper for publication in Nature Communications.

Reviewer #2

(Remarks to the Author)

It is believed that this work has found interesting and exotic domain structures in BiFeO₃ nanodisks. In my opinion, it is more appropriate to call them nested concentric radial vortices than skyrmionium ($n\pi$ -skyrmion). In fact, for the Néel-type $n\pi$ -skyrmion, the most important characteristic is that the polarization rotates continuously and radially from the core region to the peripheral region, with a phase difference of $n\pi$ (whereas n is an integer). However, the reported domain structures in this work do not satisfy this characteristic of skyrmionium.

The fatal point is that, for a typical $n\pi$ -skyrmion, its polarization rotates continuously and radially from the core region to the peripheral region with a phase difference of $n\pi$; however, for the " $n\pi$ -skyrmion" in this work, its polarization fluctuates up and down from the core region to the peripheral region (for BiFeO₃, the fluctuation angle ϕ is 109° or 71°), resulting in a phase difference of $+\phi/-\phi$ (when n is odd) or 0 (when n is even).

For example, for the reported " 2π -skyrmion" in this work, its topological charge is equal to 0 . That is because the phase difference is 0 (i.e., $+109^\circ-109^\circ=0^\circ$), instead of 2π (360°). Similarly, the " π -skyrmion" and " 3π -skyrmion" reported in the work have the same phase difference ϕ and non-integer topological charge. In BiFeO₃, ϕ may be 109° or 71° (instead of 180°), causing the topological charge being non-integer and deviating significantly from 1 (e.g., 0.648 or 0.707). However, the topological charge of a skyrmion should be strictly equal to 1 or -1 , otherwise it cannot be called a skyrmion. Therefore, the exotic structures reported in this work cannot be called skyrmion or skyrmionium ($n\pi$ -skyrmion). In addition, for the case in Fig. R2.1, the phase difference of (a), (b), (c), (d) is ϕ , $-\phi$, 0 , $-\phi$, respectively. In fact, the "skyrmionium" observed in this work is not a true skyrmionium. Strongly suggest the authors carefully review these two articles: [doi: 10.1038/s41566-023-01325-7; doi: 10.1021/acs.nanolett.7b04537], it can be confirmed that the skyrmionium from the top view (in-plane polarization component) is not the same as that in Fig.2 (l).

Overall, I think this is a good job, but it may not be suitable for publication in Nature Communications. I suggest the authors carefully consider not telling this story from the perspective of $n\pi$ -skyrmion to avoid errors.

Reviewer #3

(Remarks to the Author)

The authors have addressed all my comments adequately.

Version 2:

Reviewer comments:

Reviewer #2

(Remarks to the Author)

Reply to referees' questions and comments:

Ref number: NCOMMS-24-35844

Title: Observation of polar skyrmionium and topological transition between $n\pi$ -skyrmions

Authors: W. R. Geng, X. W. Guo, Y. L. Zhu, D. Ma, Y. L. Tang, Y. J. Wang, Y. Wu, Z. Hong, X. L. Ma

27 December, 2024

Reply to referee #1 (R1):

We appreciate the positive comment by the referee that “Detailed PFM and TEM results unveil the existence of the $n\pi$ -skyrmion in nanoislands with different sizes, and the transition between different $n\pi$ -skyrmion under external electric field” and “the work contributes an interesting and important discovery. Therefore, I recommend the publication of this manuscript after addressing the following questions”.

In the meanwhile, the referee also suggests several specific questions and comments which are summarized into five aspects. We fully understand the referee's concerns, and here we have addressed all the questions and discussed all the comments one by one in the following. The revisions are written in **RED** in the revised manuscript and the revised Supplementary Materials.

Question and comment (R1.1): One of the core findings of this study is the modulation of $n\pi$ -skyrmions by tuning the size of the islands. Is the corresponding film thickness the same? Could you provide more details on the film growth methods used? It is strongly recommended to include corresponding morphology images, as morphology significantly affects PFM phase and amplitude from the perspective of PFM testing.

Reply to Question and comment (R1.1):

We appreciate the good question and constructive suggestion from the referee. As highlighted by the referee, we realized the modulation of $n\pi$ -skyrmions by adjusting the size of the islands. The islands with different sizes were mainly stabilized in the films with different thicknesses. To make it clearly, we have summarized it in Table R1.1 to elucidate the relationships among the polar topological states, island diameters and film thicknesses, where the ‘SMR’ represents the surrounding matrix region around the nanoisland (defined in Fig. R1.1).

Table R1.1| The sizes of the nanostructures for different polar topological states.

Types	π -skyrmion	π -skyrmion	2π -skyrmion	3π -skyrmion
Diameter (nm)	100	200	350	400
Island height (nm)	3	4.5	14.5	14.5
SMR height (nm)	-	2.5	11.5	11.5

Fig. R1.1| Schematic showing the size definitions of one nanoisland.

For the methods of the film growth, we have added more details in the Method part in the revised Manuscript.

“Film deposition details. Using pulsed laser deposition (PLD) with a Coherent ComPex PRO 201 F KrF ($\lambda = 248$ nm) excimer laser, a series of epitaxial BFO thin films on LSAT (001) substrates were deposited. The LSAT (001) substrates used here are commercial substrates without extra chemical or heat treatment. Before deposition, the substrates were dipped in the 90% alcoholic solution for 12 hours to clean the organic pollutant and dusts. The substrates were affixed in the substrate plate using the silver paint solution and transferred into the main chamber of the PLD equipment. Then the substrates were heated to 850 °C for 20 minutes to clean the substrate surfaces and then cooled slowly down to the film deposition temperature at a rate of 5 °C min⁻¹. Before growing the BFO layers, the velocity for the substrate rotation motor and the target DC rotation motor are set to be 47 deg s⁻¹ and 71 deg s⁻¹, respectively. At the same time, the raster motor mode is selected for the target carousel motor, with the maximum speed being 30 deg s⁻¹ and minimum speed being 1 deg s⁻¹. The deposition of BFO films used the 1 mol% Bi-enriched BFO target, which was pre-sputtered for 20 minutes at 850 °C to clean the surface with the shutter being closed. When growing the BFO layers, the shutter is opened, a repetition rate of 8 Hz, substrate temperature of 800 °C, oxygen partial pressure of 90 mTorr and laser energy of 370 mJ were used. The distance between target and substrate is set to be 60 cm. Under the above deposition condition, the height of the laser plume is observed to be about the 1/2 of the distance between target and substrate. After deposition, these films were annealed at 800 °C in an oxygen partial pressure of 2 Torr for 20 minutes and then cooled slowly to room temperature at a rate of 5 °C min⁻¹.”

We fully agree with the referee that the film morphology significantly affects the PFM phase and amplitude from the perspective of PFM testing. As suggested by the referee, we have provided corresponding morphology images of Figs. 3a-b, e-f, i-j, as shown in Fig. R1.2-R1.4.

Indeed, the Dual AC Resonance Tracking mode (DART) was also used to confirm the phase and amplitude results during the PFM testing, which could minimize the crosstalk between the topography and the electromechanical signals¹, thereby improve the sensitivity of the measured piezoresponse signal and accurately extract the local electromechanical properties². Thus, the influence of film morphology on PFM phase

and amplitude has been excluded.

Fig. R1.2| Topography image corresponding to the region in Fig. 3a-b.

Fig. R1.3| Topography image corresponding to the region in Fig. 3e-f.

Fig. R1.4| Topography image corresponding to the region in Fig. 3i-j.

Changes in the revised Manuscript and revised Supplementary Materials:

We have added additional details about film growth in the Method part in the revised Manuscript. In addition, we have added Table R1.1, Fig. R1.1, Fig. R1.2, Fig. R1.3 and Fig. R1.4 as Table S1, Fig. S2, Fig. S3, Fig. S4 and Fig. S5 in the revised Manuscript and revised Supplementary Materials, respectively. In addition, we have cited Table S1, Fig. S2, Fig. S3, Fig. S4 and Fig. S5 in appropriate position in the revised Manuscript.

Question and comment (R1.2): Extended Data Fig. 7 shows the elemental distribution in one nanoisland. From the figure, the central region of each nanoisland (indicated by the white dashed circle) shows significantly different elemental composition, with a lack of Bi and Fe, while Al and Sr are enriched. The former originates from the film and the latter from the substrate. Combining with the results from Fig. 1h, the central position is clearly protruded (the height of this island also differs significantly from the TEM results in Extended Data Fig. 2). This seems a bit strange. Are there nanoislands on the substrate? Or could it be that diffusion occurred between the film and the substrate? Or are there defects at the center of the nanoislands? The TEM results in the article do not show obvious defects.

Reply to Question and comment (R1.2):

We appreciate the constructive discussions and insightful comments from the referee. As noticed by the referee, the EDS results in Extended data Fig. 7 show significantly different elemental composition at the central region of each nanoisland, with a lack of Bi and Fe, while Al and Sr are enriched. The special elemental distribution arises from the atomic inter-diffusion between the film and the substrate during the film deposition and the annealing procedure due to the spontaneously formed dipole disclinations in the cores of the nanoislands³⁻⁶.

As shown in Fig.1h and Extended Data Fig. 2a, the shape of the nanoisland is illuminated via both PFM topological image and cross-sectional TEM image. It is noted that the cross-sectional TEM result in Extended Data Fig. 2a suggests that the

nanostructure consists of the flat matrix and the cylinder-like hump of the BFO, which is schematized in Fig. R1.1. The protruded part revealed in Fig. 1h is potentially derived from the additional adsorbates from air due to the charged polarization state at the core of the nanoisland (Extended data Fig. 3b), which has no contribution to the phase image during PFM testing (Fig. 1i). To illuminate it, we have added a sentence “**The hump denoted by blue ellipse in (h) is derived from the additional adsorbates from air**” in the caption of Fig. 1 to avoid misunderstanding.

In addition, as revealed in the TEM images from Extended Data Fig. 2a and Fig. R1.5, no defect is observed at the center of the nanoislands. Also, there is no nanoisland on substrates. The contrast difference in the planar-view HAADF-STEM images is derived from the element diffusion between the film and the substrate during the film deposition and the annealing procedure, as revealed in the EDS results in Extended data Fig. 7.

Fig. R1.5| HAADF-STEM results for the core of the nanoislands. a, Atomic-resolved HAADF-STEM image. **b**, Enlarged HAADF-STEM image of the red rectangle in (a). **c**, Enlarged HAADF-STEM image of the blue rectangle in (b).

Changes in the revised Manuscript and revised Supplementary Materials:

To clearly illuminate the different element distribution in the core of the nanoislands in Extended data Fig. 7, some discussions have been added in Lines 5-9, Page 6 in the revised Manuscript.

“**At the core of the nanoislands, the elemental non-stoichiometry of Bi and Fe in Extended data Fig. 6 is expected to from the atomic inter-diffusion between the film and the substrate during the film deposition and the annealing procedure due to the spontaneously formed dipole disclinations in the cores of the nanoislands^{40,42-44}**”

Also, we have added a blue mark in Fig. 1h and a sentence in the caption of Fig. 1 in the revised Manuscript.

Fig. 1 | PFM analyses of high-density BFO nanoislands. **a**, Topography of BFO (001) thin film with self-assembled nanoislands. Inset is the schematic of the BFO thin film. **b**, X-ray diffraction results showing the pseudocubic 002 peaks of the films. **c**, Reciprocal space map recorded around the LSAT 002 Bragg peaks. **d-e**, Vertical PFM amplitude (V-amp.) and vertical PFM phase (V-pha.) images of the domain patterns in the BFO film. **f**, Enlarged OOP phase image superimposed with the topography image. **g**, OOP phase image of one BFO nanoisland. **h-i**, Height and vertical PFM phase spacing profiles along the yellow line in (g). **The hump denoted by blue ellipse in (h) is derived from the additional adsorbates from air.**

Question and comment (R1.3): The authors achieved the transformation from skyrmion to NTD using an applied electric field. Can the reversible transformation between $n\pi$ -skyrmions be further investigated experimentally? After all, this is essential for truly realizing a "promising candidate of information carriers."

Reply to Question and comment (R1.3):

We thank the referee for this helpful suggestion, per the request, we have performed additional experiments to confirm the transformation behaviors after opposite electric fields being applied. As shown in Fig. R1.6, the vertical PFM phase images display the domain patterns of the as-grown state, electrically wrote by 40 V bias voltage and additional wrote by opposite voltage of -40 V. Three types of domain transition processes are highlighted by red circles, green circles and yellow circles, respectively.

Fig. R1.6| Changes of polarization patterns under external electric fields. a, Vertical PFM phase image of the initial domain pattern. **b,** Vertical PFM phase image after writing experiment with a voltage of 40 V. **c,** Vertical PFM phase image after additional writing experiment with opposite voltage of -40 V. The red, green and yellow circles denote three kinds of topological transitions.

To clearly illuminate the topological transition behaviors, the enlarged vertical PFM phase images of one nanostructure are further shown in Fig. R1.7. For the nanostructure highlighted by red circles in Fig. R1.6, the topological transition process in Fig. R1.7a-b is determined to be from 3π -skyrmion-like state (as-grown state) to NTD (non-topological domain, wrote after 40 V voltage) and finally 2π -skyrmion (wrote after another opposite voltage of -40 V). For the nanostructure highlighted by green circles in Fig. R1.6, the as-grown domain state is 2π -skyrmion. After the electrical writing experiment using the voltage of 40 V, the NTD state is obtained. Then, after another electrical writing experiment with -40 V, the polarization distribution of the whole nanostructure switches back to the 2π -skyrmion (Fig. R1.7c-d). For the third transition behavior highlighted by yellow circles, which is prevalent in Fig. R1.6, the transition process is determined from 2π -skyrmion to NTD and finally π -skyrmion-like state. In a word, by applying the opposite bias voltages at the same nanostructures, the reversible transition between $n\pi$ -skyrmion-like state and NTD is realized, as shown in Fig. R1.8, which is promising for the multistate information storage using the different

signals of topological state and non-topological state.

Fig. R1.7| Topological transition processes after electrical writing experiments. a, Enlarged vertical PFM phase images of one nanostructure highlighting by red circle in Fig. R1.6. **b,** Corresponding schematics of (a), suggesting the transition from 3π -skyrmion-like state (as-grown state) to NTD (non-topological domain, wrote after 40 V voltage) and finally 2π -skyrmion (wrote after another opposite voltage of -40 V). **c,** Enlarged vertical PFM phase images of one nanostructure highlighting by green circle in Fig. R1.6. **d,** Corresponding schematics of (c), suggesting the transition from 2π -

skyrmion to NTD and finally 2π -skyrmion. **e**, Enlarged vertical PFM phase images of one nanostructure highlighting by yellow circle in Fig. R1.6. **f**, Corresponding schematics of (e), suggesting the transition from 2π -skyrmion to NTD and finally π -skyrmion-like state.

Fig. R1.8| Schematic diagrams of the three kinds of topological transitions. a, Topological transition from 3π -skyrmion-like state to NTD and finally 2π -skyrmion. **b**, Topological transition from 2π -skyrmion to NTD and finally 2π -skyrmion. **c**, Topological transition from 2π -skyrmion to NTD and finally π -skyrmion-like state.

Changes in the revised Manuscript:

According to above results, we have updated Fig. 5 in the revised Manuscript to display the rich transition behaviors under applied electric fields.

Fig. 5| Changes of polarization patterns under external electric fields. a, Vertical PFM phase image of the initial domain pattern. **b**, Vertical PFM phase image after writing experiment with a voltage of 40 V. **c**, Vertical PFM phase image after additional writing experiment with opposite voltage of -40 V. The red, green and yellow circles denote three kinds of topological transitions. **d**, Enlarged vertical PFM phase images of

one nanostructure highlighting by red circle in (a-c). **e**, Corresponding schematics of (d), suggesting the transition from 3π -skyrmion-like state to NTD and finally 2π -skyrmion. **f**, Enlarged vertical PFM phase images of one nanostructure highlighting by green circle in (a-c). **g**, Corresponding schematics of (f), suggesting the transition from 2π -skyrmion to NTD and finally 2π -skyrmion. **h**, Enlarged vertical PFM phase images of one nanostructure highlighting by yellow circle in (a-c). **i**, Corresponding schematics of (h), suggesting the transition from 2π -skyrmion to NTD and finally π -skyrmion-like state.

In addition, the corresponding discussions about Fig. 5 have also updated in Page 9 in the revised Manuscript, and Fig. R1.8 has been added as Fig. S7.

“The vertical PFM phase images of the initial state and poled state by the *dc* bias voltages of +40V and -40V are displayed in Fig. 5a-c, indicating that the changed OOP polarization distribution for the nanostructures including the nanoislands and the SMR. To elucidate the detailed topological transitions modulated by *dc* bias electric field, three kinds of topological transitions are highlighted by red (Type 1), green (Type 2) and yellow (Type 3) circles in Fig. 5a-c. The changes of OOP polarization distribution and topological transitions are further schematized in Fig. 5d-i. For the Type 1 in Fig. 5d-e and Fig. S7a, the initial domain state of one nanostructure is 3π -skyrmion-like state. After the electrical writing experiment using the voltage bias of 40 V, the 3π -skyrmion-like state transforms into the non-topological domain (abbreviated as NTD). Then after additional stimulation of -40 V at the same region, the domain state of the nanostructure is stabilized as 2π -skyrmion. During this process, the topological charge changes from $|Q| = 0.707$ to $|Q| = 0$ and finally $|Q| = 0$ (Fig. S8a). For the Type 2 in Fig. 5f-g and Fig. S7b, the topological transition process modulated by opposite voltage bias is determined to be from 2π -skyrmion to NTD and finally switching back to 2π -skyrmion, accompanying with topological charge remaining constant ($|Q| = 0$) (Fig. S8b). For the Type 3 in Fig. 5h-i and Fig. S7c, the transition from 2π -skyrmion to NTD and finally π -skyrmion-like state is schematized, with the topological charge changing from $|Q| = 0$ to $|Q| = 0$ and finally $|Q| = 0.707$, as shown in Fig. S8c. As a result, three kinds of topological transitions could be concluded under the stimulation of opposite voltage bias. The concomitant changes of topological charges (Fig. S8) suggest the feasibility of employing the topological charges as the digital bits in the promising polar-skyrmion-based multistate nonvolatile memory devices⁴⁷.”

Question and comment (R1.4): Lines 47-49, references 11 and 12 do not appear to be correct.

Reply to Question and comment (R1.4):

We appreciate the careful editing by the referee. We have updated the correct reference list in the revised Manuscript.

Question and comment (R1.5): The method of calculating the coercive field using amplitude-voltage butterfly loops in Extended Data Fig. 9 seems unscientific, and voltage should not be used as a substitute.

Reply to Question and comment (R1.5):

We appreciate the referee's concerns. It is noted that the PFM phase-voltage loops and PFM amplitude-voltage loops are obtained simultaneously for the ferroelectric polarization switching during PFM measurements, as shown in the Extended Data Fig. 9 in the previous version of Manuscript. In these loops, the horizontal coordinate represents the external applied voltage. We fully agree with the referee that the voltage should not be used as a substitute to represent the coercive field. To facilitate the conversion from the applied voltage into the applied field, the relative sizes of the tip radius and film thickness are considered. Given that the radius of the Ti/Ir (5/20) coated tips (ASYELEC-01-R) is 25 \pm 10 nm, the highest height for the BFO film with skyrmionium is 14 nm, which is smaller than the tip radius. The local electric field in the BFO film applied by the conductive tip is expected to be uniform to some extent. Thereby, during the polarization switching process, the applied field could be obtained via dividing the external applied voltage by film thickness.

In Fig. R1.9, the horizontal coordinate has been updated as the "Applied Field (kV/cm)". The coercive fields are determined from both the PFM phase-field loops and PFM amplitude-field loops in Fig. R1.9b-d, sharing the same values. As a result, the localized polarization switching and the coercive fields for different regions, including the nanoisland core (numbered region 1), nanoisland periphery (numbered region 2) and SMR (numbered region 3), are obtained. The coercive field of nanoisland core in Fig. R1.9b is calculated as 4500 kV/cm, which is larger than that of nanoisland periphery (2264 kV/cm in Fig. R1.9c) and SMR (3500 kV/cm in Fig. R1.9d), suggesting the tougher possibility of polarization switching at the same external electric fields.

Fig. R1.9| Localized polarization switching behaviors for the different regions in one skyrmionium. **a**, Vertical PFM phase image, three highlighted regions numbered as 1, 2 and 3 representing the nanoisland core, nanoisland edge and the SMR, respectively. **b-d**, Localized PFM phase-field hysteresis loops and amplitude-field butterfly loops for the three regions numbered 1, 2 and 3 in (a).

Changes in the revised Manuscript:

We have updated the Extended data Fig. 9 using the Fig. R1.9 in the revised Manuscript (numbered as Extended data Fig. 10). In addition, we have changed the discussion part of Extended data Fig. 10 in the revised Manuscript.

In Lines 3-12, Page 9: “To illuminate the polarization switching behaviors of the novel polar skyrmioniums in Fig. 2, localized PFM phase-field hysteresis loops and amplitude-field butterfly loops are compared in Extended data Fig. 10 for three representative regions, including the nanoisland core (numbered region 1), nanoisland periphery (numbered region 2) and SMR (numbered region 3). The localized coercive fields for three regions could be obtained from PFM phase-field loops and PFM amplitude-field loops in Extended data Fig. 10b-d. The coercive field of nanoisland core in Extended data Fig. 10b is calculated as 4500 kV/cm, which is larger than that of nanoisland periphery (2264 kV/cm in Extended data Fig. 10c) and SMR (3500 kV/cm in Extended data Fig. 10d), suggesting the tougher possibility of polarization switching at the same applied field.”

Reply to referee #2 (R2):

We appreciate the positive comment by the referee that “Despite the remarkable progress in the experimental observations and theoretical study on polar skyrmion, the skyrmionium and deterministic control of target skyrmions has not been experimentally observed to date. In this manuscript, the authors report the observation of self-assembly polar skyrmionium and other target skyrmions in high-density BiFeO₃ nanostructures through boundary condition engineering” and “These results enrich the family of topological structures in ferroelectrics. Overall, the current work is interesting”.

In the meanwhile, the referee also suggests several specific questions and comments which are summarized into nine aspects. We fully understand the referee’s concerns, and here we have addressed all the questions and discussed all the comments one by one in the following. The revisions are written in RED in the revised manuscript and the revised Supplementary Materials.

Question and comment (R2.1): The magnetic skyrmion Hall effect mentioned several times is a classical dynamic effect, but the skyrmionium in this article is actually a topological structure bound in high-density self-assembled nanoislands. Suggest modifying the relevant statements.

Reply to Question and comment (R2.1):

We appreciate the constructive suggestions from the referee. As suggested by the referee, we have modified the relevant statements in the Abstract and Introduction part in the revised Manuscript.

In Lines 20-23, Page 1: “Although skyrmion, with the topological number of $Q = \pm 1$, has garnered enormous interest in both magnetic and ferroelectric materials for promising candidate of information carriers, the influence of skyrmion Hall effect makes it challenging for stable information transmission in skyrmion-based electronic devices. Skyrmionium or 2π -skyrmion, one member in the family of $n\pi$ -skyrmions, has been proposed as an effective strategy for avoiding the skyrmion Hall effect in magnetic materials, while its polar counterpart is still unexplored.” has been changed into “Novel topological states have garnered enormous interest in both magnetic and ferroelectric materials for promising candidates of next-generation information carriers. Skyrmionium or 2π -skyrmion, one member in the family of $n\pi$ -skyrmions, is prevalent in magnetic materials, while its polar counterpart is still unexplored.”

In Lines 11-17, Page 2: “Complex topological structures are generally characterized by the topological number $Q^{4,5}$. Skyrmion is an extensively-studied topological state with $Q = \pm 1$, its creation, annihilation, and manipulation have been widely investigated in the field of magnetism and spintronics⁶⁻¹⁰.” has been changed into “Complex topological structures are generally characterized by their topological charge Q , which is defined as a measure of the wrapping of spin vectors around a unit sphere^{4,5,6}. Skyrmion, known as the nanoscale swirling spin texture exhibiting nontrivial real-space topology^{2,5}, is an extensively studied topological state with $Q = \pm 1$ with a variety of exotic characteristics, including robust topological protection⁷ and self-organized lattice ordering⁸. Furthermore, its creation, annihilation, and

manipulation have also been widely investigated in the field of magnetism and spintronics⁹⁻¹³.”

In Lines 18-19, Page 2: “As the promising information transmission carriers, the moving skyrmions in confined device geometries, such as the racetrack memory¹, would suffer from the detrimental effect of skyrmion Hall effect and deviate from the desired transmission path, leading to the destruction of the skyrmions at the device edges¹². To avoid the influence of skyrmion Hall effect, a promising approach is to create a spin state with the topological number being $Q = 0$, such as the skyrmionium¹¹.”
has been changed into “In addition to the widely studied skyrmion with $Q = \pm 1$, skyrmionium^{14,15} is another type of topological state characterized by a topological charge of $Q = 0$.”

Question and comment (R2.2): The manuscript focuses on the study of $n\pi$ -skyrmion, which involves the key concept of topological charge (skyrmion number) Q . Therefore, Q needs to be clearly defined. The Q value of $n\pi$ -skyrmion should be demonstrated in both experiments and calculations, rather than being roughly assumed. Please refer to previous work (Nature communications, 2015, 6(1): 8542).

Reply to Question and comment (R2.2):

We appreciate the constructive suggestion from the referee. As noticed by the referee, the topological charge (skyrmion number) Q is the key concept to describe the $n\pi$ -skyrmion in this work. Thus, as suggested by the referee, we have added the definition of the topological charge (skyrmion number) in the revised Manuscript. The recommended reference (Nature communications, 2015, 6(1): 8542) is cited as Ref. 6 in the revised Manuscript. As reported by a previous review of magnetic skyrmions⁷, the equation of the skyrmion number is given, as shown in Equation R2.1. Thus, the Q value of the $n\pi$ -skyrmion in our work could be calculated based on the detailed polarization distribution.

$$Q = \frac{1}{4\pi} \int_0^\infty dr \int_0^{2\pi} d\varphi \frac{d\theta(r)}{dr} \frac{d\phi(\varphi)}{d\varphi} \sin\theta(r) \quad \text{R2.1}$$

On the one hand, according to the polarization distribution of the skyrmionium obtained from the cross-sectional TEM sample in Extended data Figs. 2-5, the phase difference from the core to the SMR for one nanostructure is determined to be 2π (Fig. R2.1c). As a result, the topological charge Q of the skyrmionium is calculated to be 0. Similarly, the phase differences from the core to the SMR for the nanostructures in Figs. 3d, 3h, 3l are determined to be 0.5π (Fig. R2.1a), 0.5π (Fig. R2.1b) and 2.5π (Fig. R2.1d). Thus, the corresponding topological charges are calculated as 0.707, -0.707 and -0.707 respectively.

Fig. R2.1| Phase difference for different $n\pi$ -skyrmion-like states. (a) π -skyrmion-like state. (b) π -skyrmion-like state. (c) 2π -skyrmion. (d) 3π -skyrmion-like state.

On the other hand, based on the results of phase-field simulations in the revised Fig. 4, the absolute values of the topological charges for π -skyrmion, 2π -skyrmion and 3π -skyrmion are calculated as 0.836, 0.051 and 0.648, respectively. Thus, the topological charges Q for different $n\pi$ -skyrmion-like states from both experiments and calculations are demonstrated in Fig. R2.2. It is noted that the absolute value $|Q|$ (0.051) for 2π -skyrmion from phase-field simulation is close to 0, which is consistent with the experimental result to some extent. However, for the topological states of π -skyrmion and 3π -skyrmion, the topological charges from both experiments and calculations deviate from the ideal value of 1.

Fig. R2.2| Topological charges of $n\pi$ -skyrmion-like state.

Changes in the revised Manuscript:

The definition of the topological charge has been added in Line 13, Page 2 in the revised Manuscript.

“Complex topological structures are generally characterized by the topological charge Q , which is defined as a measure of the wrapping of spin vectors around a unit sphere^{4,5,6}.”

In addition, we have updated the topological charges (skyrmion number) of the $n\pi$ -skyrmion states in the revised Manuscript. Also, the recommended reference (Nature communications, 2015, 6(1): 8542) is cited as Ref. 6 in the revised Manuscript.

Question and comment (R2.3): In Figure 3g, the phase difference is only 0.5π instead of π , so its Q cannot be 1. In fact, for cases where the phase difference between the core and peripheral polarization is less than $n\pi$, Q is not an integer n . The data analysis in the manuscript should be more rigorous.

Reply to Question and comment (R2.3):

We appreciate the kind reminder from the referee. As suggested by the referee, we have recalculated the phase differences and topological charges for the different $n\pi$ -skyrmion states from both experimental and theoretical aspects, respectively. As discussed in the above reply to question and comment (R2.2), the phase differences from the core to the SMR for the different nanostructures in Fig. 3d, Fig. 3h, Fig. 2m and Fig. 3l are determined to be 0.5π (Fig. R2.1a), 0.5π (Fig. R2.1b), 2π (Fig. R2.1c) and 2.5π (Fig. R2.1d). The corresponding skyrmion numbers are calculated as 0.707, -0.707, 0 and -0.707 respectively. In addition, based on the results of phase-field simulations in the revised Fig. 4, the absolute values of the topological charges for π -skyrmion, 2π -skyrmion and 3π -skyrmion are calculated as 0.836, 0.051 and 0.648, respectively.

Thus, the nanostructures Fig. 3d, Fig. 3h, Fig. 2m and Fig. 3l are determined as π -skyrmion-like state, π -skyrmion-like state, 2π -skyrmion (skyrmionium) and 3π -skyrmion-like state.

Changes in the revised Manuscript:

In the revised Manuscript, we have updated the statements of the phase differences and topological states for different nanostructures.

Question and comment (R2.4): To my knowledge, for skyrmions in tetragonal PTO or PZT, polarization rotation π from the core (upward) to the periphery (downward) is necessary to ensure that Q is 1. However, for BFO nanodots under misfit strain, the core and peripheral polarizations in the skyrmion-like structure are difficult to point to out-of-plane distributions, resulting in Q being a non-integer. Therefore, in the manuscript, perhaps $Q=0$ for the “ 2π -skyrmion”, but for the “ π -skyrmion” mentioned in the article,

I believe the Q value of “ π -skyrmion” is not 1 (maybe less than 1). In my opinion, it is currently impossible to prove that the structures in the manuscript are members of the skyrmion family, perhaps they can be called quasi-skyrmion or skyrmion-like structures.

Reply to Question and comment (R2.4):

We appreciate the constructive discussions and suggestions from the referee. As discussed in the above reply to question and comment (R2.2), we fully agree with the referee that the Q value for the “ π -skyrmion” mentioned in the previous version of Manuscript is not 1 but less than 1. In the revised Manuscript, we have updated the corresponding description of the phase differences of the different topological states. Besides, we have changed the “ π -skyrmion”, “ 3π -skyrmion” into “ π -skyrmion-like state” and “ 3π -skyrmion-like state”, respectively.

Question and comment (R2.5): Some typos, for example, “an unique” in Line 86 should be changed to “a unique”; Some capitalization issues, for example, “the uniform” in Line 160 should be changed to “The uniform”; Some inappropriate expressions, for example, “high performance” in Line 36 should be changed to “high-performance”, the statement in line 201-202 should be further polished, etc. Please check the text and references carefully and thoroughly.

Reply to Question and comment (R2.5):

We appreciate the careful reading and kind reminder from the referee. As suggested by the referee, we have checked the text and references thoroughly and corrected some typos.

Question and comment (R2.6): In Figure 3, d, h, and l correspond to c, g, and k, respectively, right? If so, the polarization directions of d and l are drawn incorrectly. In addition, the polarization orientation of d and h does not correspond to the schematic at the bottom right corner as well.

Reply to Question and comment (R2.6):

Thanks for the kind reminder from the referee. In the revised Manuscript, we have carefully checked the Fig. 3 and corrected these points mentioned by referee.

Fig. 3| Topological transition as a function of the sizes of nanoislands. **a-b**, V-amp. and V-pha. images of the BFO film with the averaged size of nanoislands being 100 nm. **c**, Phase spacing profile around one nanoisland in the inset of (b). **d**, Reconstructed 3D polarization pattern of π -skyrmion. The schematic at the bottom right corner showing the OOP polarization distribution around one nanoisland. **e-f**, V-amp. and V-pha. images of the BFO film with the averaged size of nanoislands being 200 nm. **g**, Phase spacing profile around one nanoisland in the inset of (f). **h**, Reconstructed 3D polarization pattern of π -skyrmion. The schematic at the bottom right corner showing the OOP polarization distribution around one nanoisland. **i-j**, V-amp. and V-pha. images of the BFO film with the averaged size of nanoislands being 400 nm. **k**, Phase spacing profile around one nanoisland in the inset of (j). **l**, Reconstructed 3D polarization pattern of 3π -skyrmion. The schematic at the bottom right corner showing the OOP polarization distribution around one nanoisland.

Question and comment (R2.7): The statements in Line 202-205 should be carefully checked. Figure 4 discusses the influence of nanoisland diameter on topological domain structures, which actually is not related to topological phase transitions. In my opinion, the topological phase transition is the transition between states with different topological charges under the action of adjustable parameters (such as temperature, external electric field, etc.). However, the size of the specific nano-island is fixed and cannot be changed.

Reply to Question and comment (R2.7):

We thank the reviewer for the careful reading and valuable comment. We agree with the reviewer's statement that the topological phase transition should be related to tunable external action parameters.

We have rephrased the statement by substituting “showing the topological phase transitions from π -skyrmion, 2π -skyrmion to 3π -skyrmion with different sizes” to “showing the stabilization of three different topological phase features, i.e., π -skyrmion-like state, 2π -skyrmion, and 3π -skyrmion-like state by varying the size of the nanostructures” in Lines 16-18, Page 8 of the revised manuscript.

Question and comment (R2.8): The size of the nano-islands has a significant effect on the domain structures. In Figure 4, the simulation used a much smaller nano-island size than the experimental one, but obtained similar results. Is this reasonable as a theoretical verification?

Reply to Question and comment (R2.8):

We thank the reviewer for this insightful comment and pointing out the size difference of the BFO nanoislands between the phase-field simulations and the experimental results. Actually, our goal from the theory side is to reveal a qualitative trend, not aiming to match the exact experimental data. Based on your comment, after careful analysis, we believe that the discrepancy can be related to multiple factors: firstly, the actual strength of the depolarization field inside the nanoisland could be weaker than the theory. The potential growth defects inside the nanoisland could partially compensate for the depolarization field; Secondly, our previous model didn't consider the surrounding matrix region (SMR), which could also exaggerate the depolarization effect. To address this, we have doubled the model size in the revised simulations and added the SMR in the model (Fig. R2.3), including both in-plane and out-of-plane dimensions. The size-dependent trends in topological domain characteristics remain consistent with the original results, further supporting the theoretical verification of the experimental observations (see Fig. R2.4). We will incorporate these clarifications into the revised manuscript to address the reviewer's concern.

Fig. R2.3| Schematic of phase-field model settings of BFO nanoisland system. a, the original model of the disc-shaped BFO islands fully surrounded by air. **b,** the revised model of the disc-shaped BFO islands surrounded by matrix region (SMR), with air extending beyond them.

Fig. R2.4 | Phase-field simulations of polar topological structures of BFO islands with different diameters. **a-c**, Schematics of the nanostructures including the disc-shaped BFO islands and the surrounding BFO matrix, with the diameters of the nanostructures being d , $2d$, and $3d$, respectively. **d-f**, Planar view of out-of-plane polarization and Pontryagin density distributions in BFO nanostructures with diameters of d , $2d$, and $3d$. **g-i**, The local out-of-plane polarization distributions along a horizontal line through the center of BFO nanostructures with diameters of d , $2d$, and $3d$.

Changes in the revised Manuscript:

To explain the size difference between experiments and simulations, we added the following sentence “... as shown in Fig. 4a-c. **It should be admitted that the theoretical transition size is slightly smaller than the experimental size, which can be attributed to a higher theoretical depolarization field where the perfect charge screening is assumed. However, it can be clearly seen that the trend for the size dependent transition agrees remarkably well.** With a priori ...” in the second paragraph in Page 7 of the revised manuscript.

In addition, we have updated Fig. 4 based on the revised model (Fig. R2.3b) in the revised Manuscript. Also, the Fig. R2.3b has been added as Extended data Fig. 8 in the revised Manuscript.

Question and comment (R2.9): In the phase-field model, is the surrounding matrix region (SMR) around the nanoisland considered?

Reply to Question and comment (R2.9):

We sincerely thank the reviewer for raising this important point. Based on your suggestion, we have updated the model in the revised manuscript to introduce a surrounding matrix region (SMR) adjacent to the nanoisland, with air extending beyond them (see Fig. R2.3b), aligning it more closely with the experimental conditions. Importantly, the updated model shows that the trends in the change of domain patterns and their topological features with the diameter of the BFO nanoislands are consistent with those observed in the original model (see Fig. R2.4), further validating the reliability of the results from both models.

We have updated Fig. 4 with Fig. R2.4 in the revised manuscript and added Fig. R2.3b as Extended data Fig. 9 to clearly illustrate the setup of the computational model in the revised manuscript. Additionally, we have changed the sentences “The discrete grid points of $100\Delta x \times 100\Delta y \times 80\Delta z$ with a grid spacing of 0.4 nm are used to describe the BFO system consisting of a disc-shaped nanoisland, substrate and air layer. Corresponding to the experimental results, three different sizes of BFO islands surrounded by vacuum are considered and their diameters are set to the d , $2d$, and $3d$, with d being 24 grids, respectively.” in the Methods part to the sentences “**The discrete grid points of $200\Delta x \times 200\Delta y \times 90\Delta z$ with a grid spacing of 0.4 nm are used to describe the BFO system consisting of a disc-shaped nanoisland surrounded by matrix region, substrate and air layer (see Extended data Fig. 9). Corresponding to the experimental results, three different sizes of BFO islands are considered and their diameters are set to the d , $2d$, and $3d$, with d being 48 grids, respectively.**”, and the sentence “The total height of BFO island is 15 grids inside the simulation mesh.” in the Methods part to the sentence “**The total height of BFO island is 36 grids inside the simulation mesh.**” in revised manuscript.

Changes in the revised Manuscript:

As suggested, we have updated the phase-field model. In the revised Manuscript, we have updated Fig. 4 with Fig. R2.4 and added Fig. R2.3b as Extended data Fig. 8.

Reply to referee #3 (R3):

We appreciate the meaningful discussions and positive evaluations by the referee that “Given that skyrmionium is expected to resolve the problem of vanishing skyrmions in the proposed skyrmion devices, the work seems timely, and the conclusions seem well-supported by the experiments”.

In the meanwhile, the referee also raises some questions and comments which are summarized into seven aspects. We fully understand the referee’s concerns, and here we have addressed all the questions and discussed all the comments one by one in the following. The revisions are written in **RED** in the revised manuscript and the revised Supplementary Materials.

Question and comment (R3.1): In the introduction, the authors explain that one of the main motivations for studying the $Q=0$ skyrmions or skyrmionium is to avoid the skyrmion Hall effect, which is detrimental for the skyrmion racetrack memory-related applications. On the other hand, the skyrmionium in the BFO that the authors discovered seems to be an inherent property of the nanostructures. This makes skyrmioniums difficult to move and reduces their merits as lossless information carriers in racetrack-related devices. While the authors don’t explicitly claim that the polar skyrmionium or $n\pi$ -skyrmions can be used in racetrack devices, I think this can still be considered a discrepancy to readers.

Reply to Question and comment (R3.1):

We appreciate the careful reading of the referee and fully understand the concerns from the referee. In the revised Manuscript, we have modified the relevant statements in the introduction.

In Lines 11-17, Page 2: “Complex topological structures are generally characterized by the topological number $Q^{4,5}$. Skyrmion is an extensive-studied topological state with $Q = \pm 1$, its creation, annihilation, and manipulation have been widely investigated in the field of magnetism and spintronics⁶⁻¹⁰.” **has been changed into** “Complex topological structures are generally characterized by the topological charge Q , **which is defined as a measure of the wrapping of spin vectors around a unit sphere^{4,5,6}**. Skyrmion, **as the nanoscale swirling spin texture exhibiting nontrivial real-space topology^{2,5}**, is an extensive-studied topological state with $Q = \pm 1$ **for a variety of exotic characteristics, including the robust topological protection⁷ and self-organized lattice ordering⁸**. Furthermore, its creation, annihilation, and manipulation have **also** been widely investigated in the field of magnetism and spintronics⁹⁻¹³.”

In Lines 18-19, Page 2: “As the promising information transmission carriers, the moving skyrmions in confined device geometries, such as the racetrack memory¹, would suffer from the detrimental effect of skyrmion Hall effect and deviate from the desired transmission path, leading to the destruction of the skyrmions at the device edges¹². To avoid the influence of skyrmion Hall effect, a promising approach is to create a spin state with the topological number being $Q = 0$, such as the skyrmionium¹¹.” **has been changed into** “**In addition to the widely studied skyrmion with $Q = \pm 1$, skyrmionium^{14,15} is another kind of topological state with the topological charge being**

$Q = 0.$ ”

Question and comment (R3.2): Related to the previous question. If the polar skyrmioniums are immobile, it would be great if the authors could elaborate on how to utilize polar skyrmioniums and target skyrmion family.

Reply to Question and comment (R3.2):

Thanks for the constructive suggestion from the referee. Firstly, as revealed in the present work, the various topological states, including polar skyrmioniums and target skyrmion family, have different topological charges. Under the stimulation of external electric field, the transition among different polarization states is realized (Fig. R3.1). Three kinds of domain transitions are summarized in Fig. R3.2, which is accompanied with the changes of topological charges (Fig. R3.3). It is noted that the topological charges Q of different topological states are calculated based on Equation R3.1 reported by a previous review of magnetic skyrmions⁷.

$$Q = \frac{1}{4\pi} \int_0^\infty dr \int_0^{2\pi} d\varphi \frac{d\theta(r)}{dr} \frac{d\phi(\varphi)}{d\varphi} \sin\theta(r) \quad \text{R3.1}$$

Even though the topological states in the nanostructures are immobile, the topological charges could be designed as the “digital bits” in the potential nanoelectronics prototype devices, such as the multistate nonvolatile memory devices, which have been systematically illuminated in the recent work⁸.

Secondly, several recent works have shown that the polar skyrmions could be mobile⁹⁻¹¹. In principle, the polar skyrmionium and other polar skyrmion family should also be mobile.

Fig. R3.1| Changes of polarization patterns under external electric fields. a,

Vertical PFM phase image of the initial domain pattern. **b**, Vertical PFM phase image after writing experiment with a voltage of 40 V. **c**, Vertical PFM phase image after additional writing experiment with opposite voltage of -40 V. The red, green and yellow circles denote three kinds of topological transitions. **d**, Enlarged vertical PFM phase images of one nanostructure highlighting by red circle in (a-c). **e**, Corresponding schematics of (d), suggesting the transition from 3π -skyrmion-like state to NTD and finally 2π -skyrmion. **f**, Enlarged vertical PFM phase images of one nanostructure highlighting by green circle in (a-c). **g**, Corresponding schematics of (f), suggesting the transition from 2π -skyrmion to NTD and finally 2π -skyrmion. **h**, Enlarged vertical PFM phase images of one nanostructure highlighting by yellow circle in (a-c). **i**, Corresponding schematics of (h), suggesting the transition from 2π -skyrmion to NTD and finally π -skyrmion-like state.

Fig. R3.2| Schematic diagrams of the three kinds of topological transitions. a, Topological transition from 3π -skyrmion-like state to NTD and finally 2π -skyrmion. **b**, Topological transition from 2π -skyrmion to NTD and finally 2π -skyrmion. **c**, Topological transition from 2π -skyrmion to NTD and finally π -skyrmion-like state.

Fig. R3.3| Changes of the topological charges for three kinds of topological transitions based on experimental results. a, Topological transition from 3π -skyrmion-like state to NTD and finally 2π -skyrmion with the decreased topological charge. **b**, Topological transition from 2π -skyrmion to NTD and finally 2π -skyrmion with the topological charge remain unchanged. **c**, Topological transition from 2π -skyrmion to NTD and finally π -skyrmion-like state with the increased topological charge. |Q| denoting the absolute value of the topological charge.

Changes in the revised Manuscript and Supplementary Materials:

We have added some discussions elaborating on how to utilize polar skyrmioniums

and target skyrmion family as potential electronic devices in Lines 4-6, Page 10 in the revised Manuscript.

“The concomitant changes of topological charges (Fig. S8) suggest the feasibility of employing the topological charges as the digital bits in the promising polar-skyrmion-based multistate nonvolatile memory devices⁴⁷.”

In addition, we have added Fig. R3.2-R3.3 as Fig. S7-S8 in the revised Supplementary Materials.

Question and comment (R3.3): The authors explained that the film is under large compressive strain, and the rhombohedral and tetragonal phases coexist. Is there a connection between this and the formation of the nanoislands, or is the mixture of phases uniform?

Reply to Question and comment (R3.3):

We appreciate the constructive discussions from the referee. As noticed by the referee, the rhombohedral and tetragonal phases coexist in the BFO films under the considerable large compressive strain. As reported previously (Ma J. et al. *Nature Nanotechnology*, **13**, 947-952 (2018)), in the BFO/LSMO/LAO (001) films with large compressive strain, the coexistence of rhombohedral and tetragonal phases in the BFO films contributes much to the formation of BFO nanoislands, where the rhombohedral-phase BFO nanoislands grow from the bottom of the film and tetragonal-phase acts as the thin matrix surrounding the nanoislands. However, in our present work, the misfit strain imposed on BFO films by LSAT substrates is rather smaller than that of LAO substrates. The rhombohedral and tetragonal phases are expected to be uniform in the BFO films, which is different from the case in previous report (Ma J. et al. *Nature Nanotechnology*, **13**, 947-952 (2018)).

Changes in the revised Manuscript:

The previous work (Ma J. et al. *Nature Nanotechnology*, **13**, 947-952 (2018)) has been cited as **Ref. 28** in the revised Manuscript.

Question and comment (R3.4): It would be better if the creation mechanism for the skyrmionium is explained in more detail. While the authors reference theoretical works or the balancing mechanism between electrostatic energy and elemental non-stoichiometry, it is still difficult to see the full extent of components that determine the formation of the target skyrmions and their internal structures.

Reply to Question and comment (R3.4):

We appreciate the constructive suggestion by the referee. For the formation mechanisms of the skyrmionium, the charge accumulation, strain relaxation and depolarization field should be considered.

On the one hand, the charge accumulation plays the crucial role. The BFO films were fabricated at the high speed of 8 Hz, which facilitated to the island growth of the deposited layer. As reported previously^{3,12}, the formation of nanoislands contributes to the center-convergent or center-divergent polarization state along in-plane direction, which is consistent with our results in the Manuscript. As revealed in Fig. 2e-j in the Manuscript, the in-plane polarization of the skyrmionium is in the distribution of center-divergent state, forming the tail-to-tail charged domain walls at the nanoisland core. It is worthwhile to note that the center-divergent polarization distribution along in-plane direction is stabilized by the accumulation of particular charge, which is derived from the elemental non-stoichiometry of Bi and Fe at the nanoisland core, oxygen vacancies or other potential charged carriers.

On the other hand, the coupling between strain relaxation and depolarization field leads to the alternative polarization distribution along out-of-plane direction. According to our PFM experimental results, the skyrmionium is stabilized in 350 nm nanoislands in the 14.5 nm-thick BFO/LSAT (001) films. The lattice mismatch between BFO film ($a_{pc} = 0.3965$ nm)¹³ and LSAT substrate ($a = 0.3868$ nm)¹⁴ is of about -2.446%. To release the mismatch strain and decrease the depolarization field in the BFO films, the ferroelastic domains with alternative out-of-plane polarization are expected. It is noted that the core of the nanoislands tends to act as the preferred nucleation sites of the ferroelastic domains due to elemental non-stoichiometry, oxygen vacancies or other potential charged carriers. As a result, the skyrmionium with alternative out-of-plane polarization and center-divergent in-plane polarization is stabilized.

To further gain physical insights into the creation mechanism for these target skyrmions, different energy contributions within the 1π -, 2π - and 3π -skyrmion textures in BFO nanostructures were investigated from phase-field simulation. For simplicity, the BFO nanodisk diameter was fixed at $3d$ to analyze the individual energy density differences among pre-designed 1π -, 2π -skyrmions and the stable 3π -skyrmion structures. As shown in Figure R3.4, increasing the skyrmion order from 1π to 3π introduces more domain walls, raising the gradient energy. However, this is effectively compensated by reductions in Landau, electrostatic, and elastic energies, which relieve system strain and efficiently screen the depolarization field, allowing the 3π -skyrmion to stabilize. Therefore, from a theoretical perspective, the Landau energy, elastic energy, and electrostatic energy could serve as the driving forces for $n\pi$ -skyrmion formation in BFO nanostructures.

Fig. R3.4| Energetics comparison of the 1π -, 2π - and 3π -skyrmion textures in BFO nanostructures with diameters of $3d$ calculated from phase-field simulation. For clarity, the energy densities of the 1π -skyrmion were taken as the reference.

Changes in the revised Manuscript and Supplementary Materials:

To clearly illuminate the formation mechanism of the skyrmionium, we have added Fig. R3.4 as Extended data Fig. 9 in the revised Manuscript and updated the discussion of the formation reasons in Page 6 and Page 8 in the revised Manuscript from experimental and theoretical aspects, respectively.

Page 6: “..., which is observed prevalently in BFO film. The formation of this novel topological state could be attributed to the combined contributions of depolarization field, strain relaxation and charge accumulation. On the one hand, the BFO films fabricated via high deposition flux tend to form the nanoislands^{37,40}. At the core of the nanoislands, the elemental non-stoichiometry of Bi and Fe in Extended data Fig. 6 is expected to form the atomic inter-diffusion between the film and the substrate during the film deposition and the annealing procedure due to the spontaneously formed dipole disclinations in the cores of the nanoislands^{40,42-44}. Thus, the charge is accumulated due to the elemental difference (Extended data Fig. 6), oxygen vacancies or other potential charged carriers^{45,46}, thereby stabilizing the tail-to-tail charged domain walls and forming the center-divergent polarization along in-plane direction (Fig. 2e-j). On the other hand, the coupling effects between the depolarization field and the lattice mismatch strain at heterointerface in the BFO films stabilize the alternative out-of-plane polarization distribution, with the nanoisland cores as the preferred nucleation sites of ferroelastic domains. As a result, the skyrmionium with alternative out-of-plane polarization and center-divergent in-plane polarization is obtained in the BFO films.”

Page 8: “To further gain physical insights into the creation mechanism for these target skyrmions, different energy contributions within the 1π -, 2π - and 3π -skyrmion textures in BFO nanostructures were investigated from phase-field simulation. For simplicity, the BFO nanodisk diameter was fixed at $3d$ to analyze the individual energy density differences among pre-designed 1π -, 2π -skyrmions and the stable 3π -skyrmion structures. As shown in Extended data Fig. 9, increasing the skyrmion order from 1π to 3π introduces more domain walls, raising the gradient energy. However, this is effectively compensated by reductions in Landau, electrostatic, and elastic energies, which relieve system strain and efficiently screen the depolarization field, allowing the 3π -skyrmion to stabilize. Therefore, from a theoretical perspective, the Landau energy, elastic energy, and electrostatic energy could serve as the driving forces for $n\pi$ -skyrmion formation in BFO nanostructures.”

Question and comment (R3.5): Also, regarding the formation mechanism, does the magnetic state of the synthesized BFO play any role? What is the magnetic phase of synthesized BFO, and do they affect the formation of proposed textures?

Reply to Question and comment (R3.5):

We appreciate the constructive discussions and good questions from the referee. As noticed by the referee, BFO is one typical example of the perovskite-typed multiferroic materials, with the simultaneous existence of antiferromagnetic and ferroelectric orders¹⁵. In addition, the weak ferromagnetism can be observed at room temperature due to a residual moment in a canted spin structure¹⁶. In the present work, the magnetic state of the synthesized BFO is determined as weak ferromagnetism with the magnitude of the magnetization at the order of 10^{-3} (Fig. R3.5), thereby leading to no detected signal of magnetic domains using the magnetic force microscopy. Thus, the magnetic phase of synthesized BFO is expected to have negligible influence on the formation of ferroelectric topological states.

Fig. R3.5| Magnetic property of the BFO/LAST (001) film. (a) In-plane magnetization-magnetic field (M-H) curve of the BFO film measured at 300 K. (b) Out-of-plane M-H curve measured at 300 K.

Question and comment (R3.6): In the latter part of the manuscript, the authors use different bias voltages to control the topological number. While this suggests a possibility of controlling topological numbers, the application of voltage always seems to work in the direction where the number is decreased or maintained. Is increasing the topological number also possible, for instance, by applying reverse voltage or some additional process?

Reply to Question and comment (R3.6):

We appreciate the concerns raised by the referee. As suggested by the referee, we have performed additional experiments to confirm the transformation behaviors after opposite electric field being applied. We have found that it is possible to increase the topological number. As shown in Fig. R3.6a-c, the vertical PFM phase images display the domain patterns of the as-grown state, electrically wrote by 40 V bias voltage and additional wrote by opposite voltage of -40 V, respectively. Three kinds of domain transition processes are highlighted by red circles, green circles and yellow circles, respectively.

Fig. R3.6| Changes of polarization patterns under external electric fields. a, Vertical PFM phase image of the initial domain pattern. **b,** Vertical PFM phase image after writing experiment with a voltage of 40 V. **c,** Vertical PFM phase image after additional writing experiment with opposite voltage of -40 V. The red, green and yellow

circles denote three kinds of topological transitions.

To clearly illuminate the topological transition behaviors, the enlarged vertical PFM phase images of one nanostructure are further shown in Fig. R3.7. For the nanostructure highlighted by red circles in Fig. R3.6, the topological transition process in Fig. R3.7a-b is determined to be from 3π -skyrmion-like state (as-grown state) to NTD (non-topological domain, wrote after 40 V voltage) and finally 2π -skyrmion (wrote after another opposite voltage of -40 V). For the nanostructure highlighted by green circles in Fig. R3.6, the as-grown domain state is 2π -skyrmion. After the electrical writing experiment using the voltage of 40 V, the NTD state is obtained. Then, after another electrical writing experiment with -40 V, the polarization distribution of the whole nanostructure switches back to the 2π -skyrmion (Fig. R3.7c-d). For the third transition behavior highlighted by yellow circles, which is prevalent in Fig. R3.6, the transition process is determined from 2π -skyrmion to NTD and finally π -skyrmion-like state (Fig. R3.7e-f). In a word, by applying the opposite bias voltages at the same nanostructures, three kinds of topological transition behaviors are realized, as further shown in Fig. R3.8. Meanwhile, the changes of the topological charges accompanying with the topological transition behaviors display the different processes, including decreased, unchanged and increased trends (Fig. R3.9). Thus, it is possible to increase the topological charge by applying reverse voltage.

Fig. R3.7| Topological transition processes after electrical writing experiments. a, Enlarged vertical PFM phase images of one nanostructure highlighting by red circle in Fig. R3.5. **b,** Corresponding schematics of (a), suggesting the transition from 3π -skyrmion-like state (as-grown state) to NTD (non-topological domain, wrote after 40 V voltage) and finally 2π -skyrmion (wrote after another opposite voltage of -40 V). **c,** Enlarged vertical PFM phase images of one nanostructure highlighting by green circle in Fig. R3.5. **d,** Corresponding schematics of (c), suggesting the transition from 2π -skyrmion (as-grown state) to NTD (wrote after 40 V voltage) and finally 2π -skyrmion (wrote after another opposite voltage of -40 V). **e,** Enlarged vertical PFM phase images

of one nanostructure highlighting by yellow circle in Fig. R3.5. **f**, Corresponding schematics of (e), suggesting the transition from 2π -skyrmion (as-grown state) to NTD (non-topological domain, wrote after 40 V voltage) and finally π -skyrmion-like state (wrote after another opposite voltage of -40 V).

Fig. R3.8| Schematic diagrams of the three kinds of topological transitions. a, Topological transition from 3π -skyrmion-like state to NTD and finally 2π -skyrmion. **b**, Topological transition from 2π -skyrmion to NTD and finally 2π -skyrmion. **c**, Topological transition from 2π -skyrmion to NTD and finally π -skyrmion-like state.

Fig. R3.9| Changes of the topological charges for three kinds of topological transitions. a, Topological transition from 3π -skyrmion-like state to NTD and finally 2π -skyrmion with the decreased topological charge. **b**, Topological transition from 2π -skyrmion to NTD and finally 2π -skyrmion with the topological charge remain unchanged. **c**, Topological transition from 2π -skyrmion to NTD and finally π -skyrmion-like state with the increased topological charge. $|Q|$ denoting the absolute value of the topological charge.

Changes in the revised Supplementary Materials:

To illuminate the changes of topological charges during the topological transition processes, we have added Fig. R3.8-R3.9 as Fig. S7-S8 in the revised Supplementary Materials.

Question and comment (R3.7): The vertical PFM phase image in Figure 5 suggests that a mixture of 2π and 3π skyrmions can be initialized. Is this because the size of the nanostructure in the same film is non-uniform, or does this also have to do with the

metastability of the polar skyrmions? If such metastability is essential for determining topological numbers, analyzing the statistical distribution of skyrmions with different topological numbers might also provide insight into the energetics of skyrmion formation.

Reply to Question and comment (R3.7):

We appreciate the constructive suggestions from the referee. As noticed by the referee, there is a mixture of the 2π - and 3π -skyrmions in the same BFO film in Fig. 5a. The coexistence of 2π - and 3π -skyrmions in the same BFO film is attributed to the non-uniform sizes of the nanostructures. To clearly illuminate it, we have made a statistic about the relationship between topological states and averaged sizes of the nanostructures, as shown in Table R3.1.

Table R3.1| The sizes of the nanostructures for different polar topological states.

Types	π -skyrmion	π -skyrmion	2π -skyrmion	3π -skyrmion
Diameter (nm)	100	200	350	400
Island height (nm)	3	4.5	14.5	14.5
SMR height (nm)	-	2.5	11.5	11.5

Changes in the revised Supplementary Materials:

We have added Table R3.1 as Table S1 in the revised Supplementary Materials.

References

1. B. J. Rodriguez, C. Callahan, S. V. Kalinin, R. Proksch. Dual-frequency resonance-tracking atomic force microscopy. *Nanotechnology* **18**, 162-193 (2012).
2. N. Liu, R. Dittmer, R. W. Stark, C. Dietz. Visualization of polar nanoregions in lead-free relaxors via piezoresponse force microscopy in torsional dual ac resonance tracking mode. *Nanoscale* **7**, 11787-11796 (2015).
3. M. J. Han, Y. J. Wang, Y. L. Tang, Y. L. Zhu, J. Y. Ma, W. R. Geng, M. J. Zou, Y. P. Feng, N. B. Zhang, X. L. Ma. Shape and surface charge modulation of topological domains in oxide multiferroics. *J. Phys. Chem. C* **123**, 2557-2564 (2019).
4. M. J. Han, Y. L. Tang, Y. J. Wang, Y. L. Zhu, J. Y. Ma, W. R. Geng, Y. P. Feng, M. J. Zou, N. B. Zhang, X. L. Ma. Charged domain wall modulation of resistive switching with large on/off ratios in high density BiFeO₃ nano-islands. *Acta Mater.* **187**, 12-18 (2020).
5. V. Maurice, G. Despert, S. Zanna, M. P. Bacos, P. Marcus. Self-assembling of atomic vacancies at an oxide/intermetallic alloy interface. *Nat. Mater.* **3**, 687-691 (2004).
6. L. Crosby, J. Enterkin, F. Rabuffetti, K. Poeppelmeier, L. Marks. Wulff shape of strontium titanate nanocuboids. *Surf. Sci.* **632**, L22-L25 (2015).
7. N. Nagaosa, Y. Tokura. Topological properties and dynamics of magnetic skyrmions. *Nat. Nanotechnol.* **8**, 899-911 (2013).
8. G. Du, L. Zhou, Y. Huang, Y. Wu, H. Tian, Z. Hong. Design of polar skyrmion-based nanoelectronic prototype devices with phase-field simulations. *Adv. Funct. Mater.* 2405594 (2024).
9. L. Hu, Y. Wu, Y. Huang, H. Tian, Z. Hong. Dynamic motion of polar skyrmions in oxide heterostructures. *Nano Lett.* **23**, 11353-11359 (2024).
10. L. Hu, Y. Huang, Y. Wu, Z. Hong. Quantifying the polar skyrmion motion barrier in an oxide heterostructure. 2024. Advance Article: doi.org/10.1039/D4NR03686G
11. S. Prokhorenko, Y. Nahas, V. Govinden, Q. Zhang, N. Valanoor, L. Bellaiche. Motion and teleportation of polar bubbles in low-dimensional ferroelectrics. *Nat. Commun.* **15**, 412 (2024).
12. Zhongwen, Li, Yujia, Wang, Guo, Tian, Peilian, Lina, Zhao, Fengyuan. High-density array of ferroelectric nanodots with robust and reversibly switchable topological domain states. *Sci. Adv.* **3**, e1700919 (2017).
13. J. R. Teague, R. Gerson, W. J. James. Dielectric hysteresis in single crystal BiFeO₃. *Solid State Commun.* **8**, 1073-1074 (1970).
14. D. Mateika, H. Kohler, H. Laudan, E. Völkel. Mixed-perovskite substrates for high-T_c superconductors. *J. Cryst. Growth* **109**, 447-456 (1991).
15. T. Zhao, A. Scholl, F. Zavaliche, K. Lee, M. Barry, A. Doran, M. Cruz, Y. Chu, C. Ederer, N. Spaldin. Electrical control of antiferromagnetic domains in multiferroic BiFeO₃ films at room temperature. *Nat. Mater.* **5**, 823-829 (2006).
16. K. Wang, J.-M. Liu, Z. Ren. Multiferroicity: The coupling between magnetic and polarization orders. *Adv. Phys.* **58**, 321-448 (2009).

Reply to referees' questions and comments:

Ref number: NCOMMS-24-35844A

Title: Observation of multi-order polar skyrmion-like states and their topological transition

Authors: W. R. Geng, X. W. Guo, Y. L. Zhu, D. Ma, Y. L. Tang, Y. J. Wang, Y. Wu, Z. Hong, X. L. Ma

13 February, 2025

Reply to referee #2 (R2):

Question and comment (Reviewer #2): It is believed that this work has found interesting and exotic domain structures in BiFeO₃ nanodisks. In my opinion, it is more appropriate to call them nested concentric radial vortices than skyrmionium ($n\pi$ -skyrmion). In fact, for the Néel-type $n\pi$ -skyrmion, the most important characteristic is that the polarization rotates continuously and radially from the core region to the peripheral region, with a phase difference of $n\pi$ (whereas n is an integer). However, the reported domain structures in this work do not satisfy this characteristic of skyrmionium. The fatal point is that, for a typical $n\pi$ -skyrmion, its polarization rotates continuously and radially from the core region to the peripheral region with a phase difference of $n\pi$; however, for the “ $n\pi$ -skyrmion” in this work, its polarization fluctuates up and down from the core region to the peripheral region (for BiFeO₃, the fluctuation angle φ is 109° or 71°), resulting in a phase difference of $+\varphi/-\varphi$ (when n is odd) or 0 (when n is even).

For example, for the reported “ 2π -skyrmion” in this work, its topological charge is equal to 0 . That is because the phase difference is 0 (i.e., $+109^\circ-109^\circ=0^\circ$), instead of 2π (360°). Similarly, the “ π -skyrmion” and “ 3π -skyrmion” reported in the work have the same phase difference φ and non-integer topological charge. In BiFeO₃, φ may be 109° or 71° (instead of 180°), causing the topological charge being non-integer and deviating significantly from 1 (e.g., 0.648 or 0.707). However, the topological charge of a skyrmion should be strictly equal to 1 or -1 , otherwise it cannot be called a skyrmion. Therefore, the exotic structures reported in this work cannot be called skyrmion or skyrmionium ($n\pi$ -skyrmion). In addition, for the case in Fig. R2.1, the phase difference of (a), (b), (c), (d) is φ , $-\varphi$, 0 , $-\varphi$, respectively. In fact, the “skyrmionium” observed in this work is not a true skyrmionium. Strongly suggest the authors carefully review these two articles: [doi: 10.1038/s41566-023-01325-7; doi: 10.1021/acs.nanolett.7b04537], it can be confirmed that the skyrmionium from the top view (in-plane polarization component) is not the same as that in Fig.2 (l).

Overall, I think this is a good job, but it may not be suitable for publication in Nature Communications. I suggest the authors carefully consider not telling this story from the perspective of $n\pi$ -skyrmion to avoid errors.

Reply to Question and comment:

We fully understand the concerns from the referee and appreciate the suggestions of the more precise definition about these ferroelectric topological structures. As discussed by the referee, we agree that the definition of these ferroelectric topological structures should consider their non-integer topological charges. As noticed by the referee, it appears that the projected polarizations in a large proportion of areas in these topological structures are along $[01\bar{1}]$ and $[011]$ directions with the phase difference being 109° or 71° , which is intrinsically determined by the ferroelectric nature of spontaneous polarization and structure symmetry of BiFeO_3 .

However, it is noted that the polarization in these topological structures actually rotates continuously between neighboring regions with alternative out-of-plane polarizations, as shown in Fig. R1 (also seen as Extended data Fig. 4 in the Manuscript). The feature of continuous polarization variation is definitely different from that in well-known 109° domain walls. As shown in Fig. R2, the polarization across the 109° domain wall is sharply changed within one unit cell, which is not continuous. The difference of the polarization distribution in the two cases is also displayed as the histograms of the polarization angles in Fig. R3. For the case of 109° domain wall, the azimuth angles of polarization mainly concentrate in two angles, distributed as the manner of Gaussian to some extent (Fig. R3a), which are corresponding to the domains on both sides of the 109° domain wall (Fig. R3b). However, for the polarization from nanoisland core to nanoisland edge in Fig. 1b, the azimuth angle of polarization is diffusively distributed from -100° to 50° (Fig. R3c), suggesting the gradual polarization direction (Fig. R3d). Thus, the continuous polarization rotation bridges the alternative out-of-plane polarization in the topological states in BFO nanostructures.

Fig. R1| Polarization transition from nanoisland core to nanoisland edge. **a**, Cross-sectional atomic-resolved HAADF-STEM image. **b**, Corresponding polarization vector map, displaying the polarization transition from downward direction to upward direction.

Fig. R2| 109° domain wall in BiFeO₃ films. **a**, Atomic-resolved HAADF-STEM image. Dotted yellow line denoting the 109° domain wall. **b**, Corresponding reversed Fe-ionic displacement ($-\delta_{\text{Fe}}$) vector map.

Fig. R3| Histograms of polarization distribution. a, Distribution of the polarization angle in Fig. R2b. **b**, Schematic of the polarization across the 109° domain wall. **c**, Distribution of the polarization angle in Fig. R1b. **d**, Schematic of the polarization from nanoisland core to nanoisland edge.

Besides, it is worthwhile noting that the alternative out-of-plane polarization distribution in the BiFeO_3 nanostructures (Figs. 1e-f, and 1i in our Manuscript) is reminiscent of the distribution of magnetization direction in isolated skyrmionium in the previous paper suggested by the referee (doi: 10.1021/acs.nanolett.7b04537), with the comparison being shown in Fig. R4. Both the magnetic and ferroelectric topological structures display the doughnut-like contrast in Fig. R4. Also, the PFM results for the BiFeO_3 nanostructures (Fig. 2a-d in our Manuscript) display the radial distribution of in-plane polarization from the core region to the peripheral region. As noted by the referee, the in-plane polarization component in Fig. 2l in our Manuscript is largely reminiscent of the deformable skyrmionium state according to the previous papers (doi: 10.1038/s41566-023-01325-7; doi: 10.1021/acs.nanolett.7b04537)

Thus, the polarization states of the ferroelectric topological structures (Figs. 1e-f, and 1i in our Manuscript) are expected to be defined as ferroelectric skyrmion-like structure to some extent.

Fig. R4| Comparison between magnetic skyrmionium and ferroelectric topological state in BiFeO_3 nanostructures. **a-b**, Magnetization direction of one isolated skyrmionium (Figure from the previous paper, doi: 10.1021/acs.nanolett.7b04537). **c-d**, Out-of-plane polarization distribution of the ferroelectric topological structure reported in our Manuscript. Dashed circle denoting the doughnut-like out-of-plane polarization distribution.

However, as mentioned by the referee, the polarization changes, from core region to peripheral region in these topological structures in our Manuscript, are not strictly calculated as $n\pi$ (whereas n is an integer), which is different from the structures reported in the two recommended papers [doi: 10.1038/s41566-023-01325-7; doi: 10.1021/acs.nanolett.7b04537]. Therefore, as suggested by the referee in the last reviewer comment, the topological structures in our manuscript can be referred as multi-order "skyrmion-like" states, more precisely described as a family of nested concentric vortices.

By further combining the referee's suggestions in this reviewer comment, we tend to tell the story by emphasizing the multi-layered nested concentric radial topological states. Thus, in the revised manuscript, the π -skyrmion, 2π -skyrmion and 3π -skyrmion has been changed as one-order skyrmion-like structure, two-order skyrmion-like

structure and three-order skyrmion-like structure. In these updated definitions, the change of nested order is highlighted, not the phase difference. According to the experimental and theoretical results in our Manuscript, the stabilization of multi-order skyrmion-like structures is realized by tuning the sizes of BiFeO₃ nanostructures, and the transition between different topological states is realized under external electric field.

Changes in the revised Manuscript:

Based on the referee's discussions and suggestions, we have changed the description of observed ferroelectric topological structures from "n π -skyrmion-like states" to "multi-order skyrmion-like states" and the "skyrmionium" to "two-order skyrmion-like structure" in the revised Manuscript.

In addition, we have added the recommended references as Ref. 16-17 in the revised Manuscript.

“16. Shen, Y. et al. Optical skyrmions and other topological quasiparticles of light. *Nat. Photonics* 18, 15-25 (2024).

17. Zhang, S. et al. Real-space observation of skyrmionium in a ferromagnet-magnetic topological insulator heterostructure. *Nano Lett.* 18, 1057-1063 (2018).”

Reply to referees' questions and comments:

Ref number: NCOMMS-24-35844B

Title: Observation of multi-order polar radial vortices and their topological transition

Authors: W. R. Geng, X. W. Guo, Y. L. Zhu, D. Ma, Y. L. Tang, Y. J. Wang, Y. Wu, Z. Hong, X. L. Ma

6 March, 2025

Reply to referee #2 (R2):

Question and comment (Reviewer #2): I noted that the paper title has been changed. Considering that the experimental data is reliable, even though the discovered domain structures in the manuscript have nothing to do with the real skyrmionium (or $n\pi$ -skyrmions), I can accept the publication of this manuscript. Nevertheless, the domain structure reported in the manuscript is not skyrmionium but a high-order radial vortex with its dipoles periodically fluctuating along radius direction. In my opinion, the authors still lack understanding of the geometric properties of skyrmionium so far. In the following, I will explain my viewpoint in detail.

(1) Firstly, the characteristics of skyrmion.

Skyrmion is a topological soliton that widely exists in magnetic, optical, liquid crystal, and other systems, and has recently been discovered in ferroelectric systems as well. There are two types of skyrmions, Bloch-type and Néel-type. To describe topological invariance, skyrmion or skyrmionium can be described by the skyrmion number (topological charge) Q . In the continuum case, the topological charge Q is defined as the number of times normalized polarization (spin) winds around the sphere S^2 . Thus, the topological number of skyrmion or skyrmionium must be integers (similarly, the topological number of meron or meronium is a semi-integer).

The geometric feature of Néel-type skyrmion is polarization (or spin) rotating 180° from the core to the periphery along radius direction, ensuring the topological charge of 1. Similarly, the geometric feature of $n\pi$ -skyrmion is polarization (or spin) rotating $n \times 180^\circ$ from the core to the periphery along radius direction, ensuring the topological charge of 0 (when n is even) or 1 (when n is odd). In this work, the domain structure involved is related to Néel-type skyrmion. However, the rotation angle of polarization from core to periphery much less than 180° (Fig.S1 A), resulting in the topological charge Q of the so-called skyrmion is less than 1 (experimental value 0.707, simulated value 0.836). In fact, it is possible

theoretically to create a standard skyrmion with $Q=1$ in BFO (Fig.S3 B). By comparison between Fig.S1 A and B, the so-called “ π -skyrmion” in this work (Fig.S3 A) is far-fetched to be identified as a skyrmion.

(2) Second, the characteristics of skyrmionium. This section is the main idea and focus of the original manuscript, but there exists significant conceptual problem.

As mentioned above, the geometric feature of $n\pi$ -skyrmion is polarization (or spin) rotating $n \times 180^\circ$ from the core to the periphery along radius direction, ensuring the topological charge of 0 (when n is even) or 1 (when n is odd). Take 2π -skyrmion as an example, the polarization continuously rotates 360° from core to periphery along radius direction, making the topological charge $Q=0$. Fig.S1 B exhibits the skyrmionium configuration from classical literatures. It can be seen that, distinct from the domain structure in this work (Fig.S1 A and Fig.S2 A), the in-plane polarization in real 2π -skyrmion does not always point towards the same direction, but instead alternates between centripetal and centrifugal directions (Fig.S1 B and Fig.S2 B).

However, the “ 2π -skyrmion” in this work (as shown in Fig.S1 A) does not possess the configuration characteristics of real 2π -skyrmion. The polarization in it does not continuously rotate 360° along the radius direction, but first rotates upwards $+\varphi^\circ$ and then downwards $-\varphi^\circ$ (with a total phase of 0°), resulting in its in-plane polarization always pointing towards the periphery direction (Fig.S1 A and Fig.S2 A). The so-called “ 2π -skyrmion” in this work is more like a radial vortex with its dipoles periodically fluctuating along the radius. Noting that, the out-of-plane polarization distribution of both the real 2π -skyrmion (skyrmionium) and the so-called “ 2π -skyrmion” reported in this work look the same (Fig.S2). Therefore, to determine skyrmionium, observing only the out-of-plane polarization distribution is evidence-insufficient, and the in-plane polarization distribution is needed as well. Fig.S2 shows that, although the “ 2π -skyrmion” claimed in this work and the real 2π -skyrmion have the same out-of-plane polarization configuration, they have completely different in-plane polarization distributions. In short, the “ 2π -skyrmion” claimed in the manuscript is not real 2π -skyrmion (skyrmionium). The same error happens in the case of so-called “ $n\pi$ -skyrmion”. In other words, although the experimental data of this work is reliable, it has nothing to do with $n\pi$ -skyrmion (skyrmionium). I sincerely suggest the authors to rewrite the article, for example from the perspective of high-order radial vortex.

Reply to Question and comment:

Thanks for the in-depth discussions from the referee about the geometric properties of skyrmionium. We fully understand the concerns from the referee and

appreciate the suggestions. We have changed the concept of “multi-order polar skyrmion-like states” into “multi-order polar radial vortices” as suggested by the referee and made some relevant changes in the revised Manuscript and Supplementary Materials.

I noted that the paper title has been changed. Considering that the experimental data is reliable, even though the discovered domain structures in the manuscript have nothing to do with the real skyrmionium (or $n\pi$ -skyrmions), I can accept the publication of this manuscript. Nevertheless, the domain structure reported in the manuscript is not skyrmionium but a high-order radial vortex with its dipoles periodically fluctuating along radius direction. In my opinion, the authors still lack understanding of the geometric properties of skyrmionium so far. In the following, I will explain my viewpoint in detail.

(1) Firstly, the characteristics of skyrmion.

Skyrmion is a topological soliton that widely exists in magnetic, optical, liquid crystal, and other systems, and has recently been discovered in ferroelectric systems as well. There are two types of skyrmions, Bloch-type and Néel-type. To describe topological invariance, skyrmion or skyrmionium can be described by the skyrmion number (topological charge) Q . In the continuum case, the topological charge Q is defined as the number of times normalized polarization (spin) winds around the sphere S^2 . Thus, the topological number of skyrmion or skyrmionium must be integers (similarly, the topological number of meron or meronium is a semi-integer).

The geometric feature of Néel-type skyrmion is polarization (or spin) rotating 180° from the core to the periphery along radius direction, ensuring the topological charge of 1. Similarly, the geometric feature of $n\pi$ -skyrmion is polarization (or spin) rotating $n \times 180^\circ$ from the core to the periphery along radius direction, ensuring the topological charge of 0 (when n is even) or 1 (when n is odd). In this work, the domain structure involved is related to Néel-type skyrmion. However, the rotation angle of polarization from core to periphery much less than 180° (Fig.S1 A), resulting in the topological charge Q of the so-called skyrmion is less than 1 (experimental value 0.707, simulated value 0.836). In fact, it is possible theoretically to create a standard skyrmion with $Q=1$ in BFO (Fig.S3 B). By comparison between Fig.S1 A and B, the so-called “ π -skyrmion” in this work (Fig.S3 A) is far-fetched to be identified as a skyrmion.

(2) Second, the characteristics of skyrmionium. This section is the main idea and focus of the original manuscript, but there exists significant conceptual problem.

As mentioned above, the geometric feature of $n\pi$ -skyrmion is polarization (or spin) rotating $n \times 180^\circ$ from the core to the periphery along radius direction, ensuring the topological charge of 0 (when n is even) or 1 (when n is odd). Take 2π -skyrmion as an example, the polarization continuously rotates 360° from core to periphery along radius direction, making the topological charge $Q=0$. Fig.S1 B exhibits the skyrmionium configuration from classical literatures. It can be seen that, distinct from the domain structure in this work (Fig.S1 A and Fig.S2 A), the in-plane polarization in real 2π -skyrmion does not always point towards the same direction, but instead alternates between centripetal and centrifugal directions (Fig.S1 B and Fig.S2 B).

However, the “ 2π -skyrmion” in this work (as shown in Fig.S1 A) does not possess the configuration characteristics of real 2π -skyrmion. The polarization in it does not continuously rotate 360° along the radius direction, but first rotates upwards $+\varphi^\circ$ and

then downwards $-\varphi^\circ$ (with a total phase of 0°), resulting in its in-plane polarization always pointing towards the periphery direction (Fig.S1 A and Fig.S2 A). The so-called “ 2π -skyrmion” in this work is more like a radial vortex with its dipoles periodically fluctuating along the radius. Noting that, the out-of-plane polarization distribution of both the real 2π -skyrmion (skyrminium) and the so-called “ 2π -skyrmion” reported in this work look the same (Fig.S2). Therefore, to determine skyrmionium, observing only the out-of-plane polarization distribution is evidence-insufficient, and the in-plane polarization distribution is needed as well. Fig.S2 shows that, although the “ 2π -skyrmion” claimed in this work and the real 2π -skyrmion have the same out-of-plane polarization configuration, they have completely different in-plane polarization distributions. In short, the “ 2π -skyrmion” claimed in the manuscript is not real 2π -skyrmion (skyrminium). The same error happens in the case of so-called “ $n\pi$ -skyrmion”. In other words, although the experimental data of this work is reliable, it has nothing to do with $n\pi$ -skyrmion (skyrminium). I sincerely suggest the authors to rewrite the article, for example from the perspective of high-order radial vortex.

Fig.S1. Configuration comparison between the “ 2π -skyrmion” claimed in this work (A) and the real 2π -skyrmion defined in classical literatures (B). It can be seen that polarization in A) rotates with finite fluctuation up and down alternatively along the radius direction, with the total phase of 0. However, the polarization in B) rotates counterclockwise continuously 360° along the radius direction. By comparison, it can be concluded that A) is not 2π -skyrmion (skyrminium) but a radial vortex with its dipoles periodically fluctuating along the radius.

Fig.S2. For the “ 2π -skyrmion” claimed in this work and the real 2π -skyrmion, they have the same out-of-plane polarization configuration, but they have completely different in-plane polarization distributions. By comparison, it can be concluded that **A**) is not 2π -skyrmion (skyrmionium).

Fig.S3. Schematics of $n\pi$ -skyrmion (skyrmionium) in BFO nanodisk. **A)** Schematic of the “ $n\pi$ -skyrmion” in this work provided in the last response. It can be seen that the dipoles rotate with finite fluctuations up ($+\varphi$) and down ($-\varphi$) alternatively along the radius direction. **B)** Schematic of the real $n\pi$ -skyrmion (skyrmionium) in BFO nanodisk. It can be seen that the dipoles rotate continuously by $n \times 180^\circ$ along the radius direction. By comparison, it can be concluded that **A)** is incorrect.